# Mycobacterial dynamin-like protein IniA mediates membrane fission

Manfu Wang[1,2,3,8], Xiangyang Guo [4,8], Xiuna Yang[1,2], Bing Zhang[1,2], Jie Ren[5], Aijun Liu[6], Yajun Ran[1,2], Bing Yan[5], Fang Chen[5], Luke W. Guddat[7], Junjie Hu [4,5], Jun Li[1,2] & Zihe Rao[1,4,5,6]

*Mycobacterium tuberculosis* infection remains a major threat to human health worldwide. Drug treatments against tuberculosis (TB) induce expression of several mycobacterial proteins, including IniA, but its structure and function remain poorly understood. Here, we report the structures of *Mycobacterium smegmatis* IniA in both the nucleotide-free and GTP-bound states. The structures reveal that IniA folds as a bacterial dynamin-like protein (BDLP) with a canonical GTPase domain followed by two helix-bundles (HBs), named Neck and Trunk. The distal end of its Trunk domain exists as a lipid-interacting (LI) loop, which binds to negatively charged lipids for membrane attachment. IniA does not form detectable nucleotide-dependent dimers in solution. However, lipid tethering indicates nucleotide-independent association of IniA on the membrane. IniA also deforms membranes and exhibits GTP-hydrolyzing dependent membrane fission. These results confirm the membrane remodeling activity of BDLP and suggest that IniA mediates TB drug-resistance through fission activity to maintain plasma membrane integrity.

[1] Shanghai Institute for Advanced Immunochemical Studies and School of Life Science and Technology, ShanghaiTech University, Shanghai 201210, China. [2] CAS Center for Excellence in Molecular Cell Science, Shanghai Institute of Biochemistry and Cell Biology, Chinese Academy of Sciences, Shanghai 200031, China. [3] University of Chinese Academy of Sciences, Beijing 100101, China. [4] State Key Laboratory of Medicinal Chemical Biology, College of Life Sciences, Nankai University, and Tianjin Key Laboratory of Protein Sciences, Tianjin 300071, China. [5] National Laboratory of Biomacromolecules, CAS Center for Excellence in Biomacromolecules, Institute of Biophysics, Chinese Academy of Science, Beijing 100101, China. [6] Laboratory of Structural Biology, School of Medicine, Tsinghua University, Beijing 100084, China. [7] School of Chemistry and Molecular Biosciences, The University of Queensland, Brisbane 4072 QLD, Australia. [8] These authors contributed equally: Manfu Wang, Xiangyang Guo. Correspondence and requests for materials should be addressed to J.H. (email: huj@ibp.ac.cn) or to J.L. (email: lijun@sibcb.ac.cn)

Tuberculosis (TB) is commonly treated with up to five frontline antibiotics including isoniazid (INH) and ethambutol (EMB), both of which inhibit mycobacterial cell wall biogenesis[1–3]. The induction of several bacterial genes conveys drug resistance, but little is known about their molecular mechanisms[4–6]. Among the drug inhibitory genes, three are clustered in one operon termed *iniBAC* (INH-inducible genes B, A, and C)[4]. IniB is homologous to a glycine-rich cell wall structural protein from *Arabidopsis thaliana*[7], which is consistent with its function in cell wall stabilization. *IniA* and *iniC* both encode GTPase domain containing proteins predicted to be homologs of bacterial dynamin-like protein (BDLP) and, therefore, members of the dynamin superfamily[8,9]. Functional investigation of these BDLPs would provide critical insight into mycobacterial adaption upon drug treatment and subsequently offer new clues toward optimization of antibiotic strategies.

The molecular function and activity of BDLPs is unclear. IniA mediates drug tolerance[7], tandem BDLPs DynA and DynB in the filamentous bacteria *Streptomyces* play a role in cytokinesis during sporulation[8,10], and LeoA in enterotoxigenic *Escherichia coli* (*E. coli*) strain H10407 is proposed to be linked to toxin secretion[11,12]. In contrast, the molecular architecture of BDLP has been well documented, which in principle is very similar to its eukaryotic counterpart dynamin. The GTPase domain is closely associated by a helix bundle (HB) type stalk structure. The helices in the stalk first exit the GTPase domain and then loop back, positioning the N- and C-termini in close proximity. The cyanobacteria BDLP forms a dimer in both *apo* and GDP-bound states[13]. The proximal HB relative to the GTPase domain (HB1 or the Neck) and the distal HB (HB2 or the Trunk) exhibit a sharp bend. A helix hairpin termed the Paddle, which is likely a transmembrane domain, is inserted in between the third and fourth helices of the Trunk. Dimerization is mediated by the Paddle, and the GTPase domain in the GDP state. When the BDLP is added to a lipid tube in the presence of GMPPNP, the protein inserts into lipids using the Paddle domain and assembles into helical filaments[13,14]. The GTPase domain maintains a dimer, but the Neck and Trunk straighten and stand up on the membrane[14]. The structure of BDLP has also been obtained with LeoA, in which the protein is a nucleotide-free monomer[11]. The GTPase domains can be superimposed with that of the cyanobacteria BDLP, and there is no bending in the stalk region.

The overall configuration of BDLP is predicted to be similar to another class of dynamin-like proteins (DLPs), the mitofusins (MFNs), which mediate outer mitochondrial membrane fusion. The structures of the minimal GTPase domain (MGD) align with the GTPase and Neck region of BDLP[15]. The predicted secondary structure of the remaining MFN resembles the Trunk and Paddle of BDLP[16]. Such evolutionary conservation between the two DLPs suggests similar molecular activity. The conformational changes observed in BDLP are referred by predicting models of the fusogenic action of MFN[15,16]. However, many other dynamin superfamily members, including dynamins themselves, also mediate membrane fission. Here, we determined structures of IniA, with and without bound GTP, and demonstrated that it mediates membrane fission in vitro. Our results suggested that IniA, like other BDLPs, has weak GTPase activity and nucleotide-dependent dimerization in solution, but tends to form nucleotide-independent homotypic interactions on the membrane and cuts the membrane. These features are linked to IniA-mediated isoniazid resistance.

## Results

**Crystal structures of IniA**. To gain insight into IniA-mediated drug resistance, we determined the structure of full-length IniA from *Mycobacterium smegmatis* (*M. smegmatis*), which shares 68% sequence identity with IniA from *Mycobacterium tuberculosis* (*M. tuberculosis*) (Supplementary Fig. 1). IniA was expressed in *E. coli*, purified, and crystallized in the presence of GTP (Fig. 1a). A 2.2 Å resolution structure was determined by the single-wavelength anomalous diffraction (SAD) method (Table 1). One IniA molecule is present in the asymmetric unit. IniA consists of an N-terminal GTPase domain and two HBs (Fig. 1b). The HB1/Neck is formed by four helices: α(−1) and α0 are N-terminal extensions of the GTPase domain, α6b follows α6a which is the last helix of the GTPase domain, and α11b is the last helix of the polypeptide. The HB2/Trunk is composed of three long helices (α7, α8, and α11a) and a parallel long loop connecting α8 and α11a. The loop corresponds to several helices in the Trunk and Paddle domains of the cyanobacteria BDLP. For ease of comparison with other DLPs, particularly MFN, we termed the region close to α8 the α9-loop and omitted α9 and α10 in the secondary structure numbering of IniA.

Notably, there is a sharp bend between HB1/Neck and HB2/Trunk of IniA, reminiscent of the configuration of the cyanobacteria BDLP in the lozenge dimer (Fig. 1c). The bending between HB1 and HB2 occurs at the α6b-α7 and α11a-α11b hinges (Supplementary Fig. 2a). In the corresponding positions of cyanobacteria BDLP, they are termed hinge 1a and hinge 1b, respectively. Both hinge 1a and hinge 1b of IniA are shaped by a cluster of charged residues (Supplementary Fig. 2a).

The relative position between the GTPase domain and Neck domain in IniA is similar to the BDLP lozenge dimer, but is different from the GMPPNP-bound BDLP and nucleotide-free LeoA (Fig. 1d; Supplementary Fig. 3b). The HB1 of IniA has little contact with the GTPase domain and adopts a conformation similar to the HB-open state of MFN1-MGD[16]. In contrast, HB2 interacts extensively with its GTPase domain. The α2 and α3 helices in the GTPase domain form a short HB with part of α8 in the HB2 (Supplementary Fig. 2b). In addition, the α9-loop runs through a groove region formed by the central β-sheet of the GTPase domain. First, F463 in the α9-loop is sandwiched by R38 and R123 in the GTPase domain; then the L159 in the GTPase domain is sandwiched by F463 and M465 in the α9-loop. Additional interactions come from neighboring hydrogen bonds (Supplementary Fig. 2b).

We also determined the structure of nucleotide-free IniA at 3.2 Å resolution by molecular replacement using the GTP-bound structure as a search model (Table 1). Overall, the *apo* state of IniA adopts a very similar fold as the GTP state, with a root-mean-square deviation of only 0.525 Å (Fig. 1c). An IniA dimer with antiparallel packing of HB2 was found in the asymmetric unit. Even though there is only a monomer present in the asymmetric unit of the GTP-bound IniA structure, through crystallographic symmetry a similar dimer is also observed in that complex (Supplementary Fig. 3a).

**Nucleotide binding and GTP hydrolysis**. The GTPase domain of IniA has a similar fold to other dynamin superfamily members. Four signature motifs wrap around GTP: G1 (P-loop), G2 (switch 1), G3 (switch 2), and G4 (Fig. 2a). As expected, K49 and S50 in the G1, T70 in G2, and D141 in G3, with the help of a magnesium ion, coordinate the binding of the phosphate moieties in the nucleotide. K200 and D202 in G4 engage the ribose. K46 in G1, equivalent to the catalytic R77 of human atlastin 1 (ATL1, another dynamin-like GTPase that mediates fusion of the endoplasmic reticulum)[17,18], points toward the bridging oxygen between the beta-phosphate and gamma-phosphate, suggesting a role in charge compensation during GTP hydrolysis. This lysine is conserved in cyanobacteria BDLP, but it does not contact the

phosphates in the GDP-bound state. Furthermore, the G2 of IniA forms a U-shaped flap on top of GTP, likely stabilizing the positions of several key residues in the catalytic core. The curving of the G2 loop is stabilized by a hydrogen bond between the side chain of R63 and the carbonyl oxygen of S69 (Fig. 2a). By comparison, the equivalent G2 loop in GDP-bound cyanobacteria BDLP is largely disordered and further away from the nucleotide (Fig. 2b). Notably, the G2 of cyanobacteria BDLP has a similar shape to that of IniA, suggesting such G2s would not interfere with dimerization of the GTPase domain.

To determine the nucleotide-binding affinity of IniA, we performed isothermal titration calorimetry (ITC). Wild-type IniA binds to GDP with a $K_d$ value of 0.75 μM and GMPPNP with a $K_d$ value of 9.01 μM (Fig. 2c). Given that GTP was captured and retained in IniA crystals, we speculated that GTP hydrolysis by IniA is extremely slow, especially at low temperature. We thus performed ITC analysis using GTP, and yielded a $K_d$ value of 0.74 μM, close to that of GDP. These results suggest that IniA interacts with nucleotides efficiently, and GMPPNP does not adequately mimic GTP for binding to IniA. When K46 was mutated to alanine, the mutant IniA largely maintained interactions with GDP, showing only a twofold reduction in affinity for GMPPNP or GTP (Fig. 2c).

Consistent with the fact that GTP is preserved during crystallization, we found that IniA is a poor GTPase, with a $k_{cat}$ value of 0.29 min$^{-1}$, which is similar to that observed for BDLP[13]. Unlike MFN1, the GTPase activity of IniA was not enhanced by the presence of potassium ion (Supplementary Fig. 4a). As expected, the K46A mutation reduced GTPase activity, and S50A and R63A, individually or in combination, drastically compromised GTP hydrolysis (Fig. 2d). In contrast, mutation of the conserved K49 residue in the G1/P-loop only marginally affected the enzymatic activity of the GTPase (Fig. 2d). Similarly, removal of the HB2/Trunk domain (ΔHB2) had little impact on the GTPase activity (Fig. 2d). It has been shown previously that the GTPase activity of dynamin-1 is drastically stimulated in the presence of membranes[19]. We thus measured IniA GTPase activity in the presence and absence of liposomes, but observed no significant increase in the presence of liposomes (Supplementary Fig. 4b).

**Lipid association.** Many dynamin superfamily members interact with membranes[20]. IniA is purified as a soluble protein, but likely binds to the membrane. Cyanobacteria BDLP uses the Paddle domain, in the form of a helix hairpin, to integrate into membranes[14]. A potential transmembrane segment (residues 463–485) was predicted by the TMHMM Server[21] in the equivalent region of IniA (Supplementary Fig. 1). Interestingly, this region appears as a loop that follows the α9-loop in our structures (Fig. 3a). To investigate the membrane association by IniA, we performed a liposome flotation assay. In this assay, liposomes are incubated with IniA and placed beneath a sucrose density gradient; after centrifugation, liposomes migrate to the top of the gradient and membrane binding is measured by co-migration of the protein. As expected, IniA efficiently bound to liposomes composed of *E. coli* polar lipids (EPL: 78 mol% phosphatidylethanolamine [PE]; 12 mol% phosphatidylglycerol [PG]; 6 mol% cardiolipin [CL]; and 4 mol% phosphatidic acid [PA]). When the NaCl concentration was increased from 150 mM to 600 mM, the interaction was drastically disrupted (Fig. 3b), suggesting a hydrophilic mode of binding. We then tested whether IniA prefers charged lipids. Consistently, IniA floated well with liposomes containing PG, CL, or PA, but did not engage PC or PC + PE liposomes (Fig. 3c). Among the chosen lipids, bacterial signature lipid CL is most effective in recruiting IniA. We also tested nucleotide dependency

**Table 1 Data collection and refinement statistics**

| | IniA (Pt derivative) | IniA (GTP-bound) | IniA (nucleotide-free) |
|---|---|---|---|
| *Data collection* | | | |
| Space group | $P6_4$ | $P6_4$ | $P3_1$ |
| Cell dimensions | | | |
| $a, b, c$ (Å) | 91.15, 91.15, 139.39 | 91.51, 91.51, 139.49 | 91.76, 91.76, 137.34 |
| $\alpha, \beta, \gamma$ (°) | 90, 90, 120 | 90, 90, 120 | 90, 90, 120 |
| Wavelength (Å) | 0.9779 | 0.9779 | 0.9785 |
| Resolution (Å) | 50.00-2.60 (2.64-2.60)$^a$ | 50.00-2.20 (2.24-2.20) | 50.00-3.20 (3.26-3.20) |
| Completeness (%) | 100.0 (100.0) | 100.0 (100.0) | 99.9 (100.0) |
| $R_{merge}$ (%) | 13.7 (73.7) | 13.1 (53.9) | 19.0 (81.3) |
| Redundancy | 10.3 (10.3) | 20.6 (21.2) | 5.2 (5.3) |
| $I/\sigma(I)$ | 19.5 (3.0) | 25.4 (4.0) | 7.4 (1.9) |
| *Refinement* | | | |
| Resolution (Å) | | 46.50-2.20 | 39.73-3.21 |
| No. of reflections | | 33,560 | 21,086 |
| $R_{work}/R_{free}$ (%) | | 18.5/22.1 | 21.9/25.8 |
| No. of atoms | | | |
| Protein | | 4343 | 8375 |
| Ligand | | 43 | |
| Water | | 333 | |
| B-factor (Å$^2$) | | | |
| Protein | | 48.92 | 55.42 |
| Ligand | | 51.93 | |
| Water | | 48.19 | |
| R.m.s deviations | | | |
| Bond lengths (Å) | | 0.004 | 0.012 |
| Bond angles (°) | | 0.844 | 2.127 |
| Ramachandran plot (%) | | | |
| Favored | | 97.1 | 97.7 |
| Allowed | | 2.7 | 2.1 |
| Outliers | | 0.2 | 0.2 |

$^a$Values in parentheses are for highest-resolution shell

of IniA–lipid interactions, and found no detectable changes of the flotation pattern when various nucleotides were added (Supplementary Fig. 4c). The S50A/R63A mutant, which fails to engage the nucleotide, interacted with liposomes, which is consistent with what occurs for the wild-type protein in the same experiment (Supplementary Fig. 4g).

Analyzing the lipid-binding mode, we found a tartaric acid (TA) molecule (from the crystallization buffer) bound next to the predicted TM region. (Fig. 3a). The negative charge of TA resembles that of the phosphate group in the hydrophilic head of phospholipids. Interestingly, when we changed the 480–492 region to a GGGGSGGGGS linker (Δ480–492), which surround the TA in the structure, we found that membrane binding was completely abolished (Fig. 3d). Thus, we termed this region the lipid-interacting (LI) loop. When hydrophobic residues, including V485 and L486, were mutated, lipid binding was only marginally affected. In contrast, when positively charged residues, particularly R488 and K489, were mutated, lipid binding was drastically reduced (Fig. 3d). The ΔHB2 mutant, which lacks the HB2/Trunk domain but possesses intact LI loop, interacted efficiently with the membrane (Supplementary Fig. 4g). These results confirm that IniA utilizes the positively charged residues in the LI loop to attach to membranes.

Next, we tested the lipid binding of IniA in a cellular context. GFP-tagged IniA was transformed into the *M. smegmatis* strain. Wild-type IniA exhibited similar localization as DynA, which is predominantly on the plasma membrane, occasionally in a punctate pattern (Fig. 3e). However, the mutant Δ480–492 was

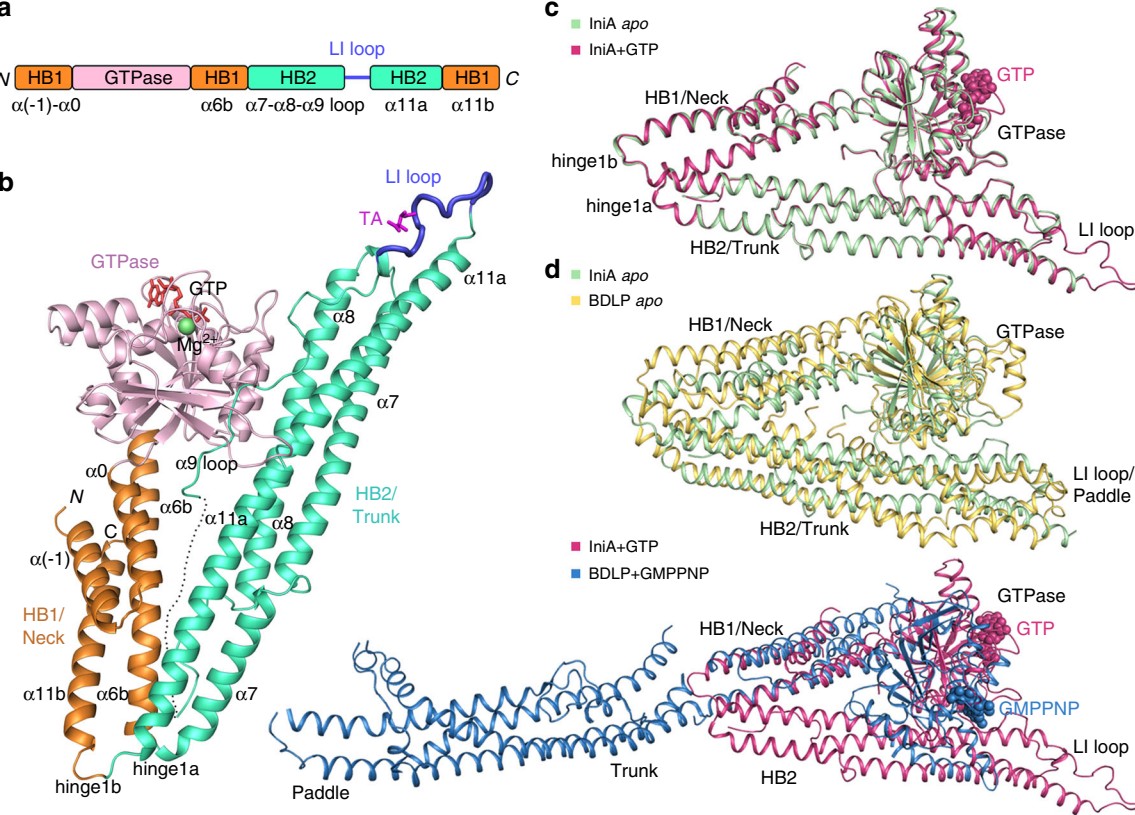

**Fig. 1** Overall structures of IniA. **a** Scheme showing the domains of IniA in different colors. Secondary structure elements that form HBs are labeled. N, N-terminus; C, C-terminus. **b** Cartoon structure of GTP-bound IniA. As in (**a**), the GTPase, HB1/Neck, and HB2/Trunk domains and LI loop are colored pink, orange, green, and blue, respectively. GTP and tartaric acid (TA) are shown as red and magenta sticks. $Mg^{2+}$ is represented by a green sphere. The disordered segment in the α9-loop is denoted by a dotted line. **c** Structural superposition of the *apo* IniA and GTP-bound IniA. GTP is shown as spheres. **d** Structural comparison of the *apo* structures between IniA and BDLP (PDB code: 2J69) (upper panel), and the GTP/GMPPNP-bound structures between IniA and BDLP (PDB code: 2W6D) (lower panel)

mostly cytosolic (Fig. 3f), consistent with a lack of membrane association. Similarly, R448D mutant was not seen on the plasma membrane (Fig. 3g). Some bright spots were seen with both mutants (Fig. 3f, g), presumably indicating that this mutant protein aggregates due to overexpression. Notably, the plasma membrane of cells transformed with wild-type IniA-GFP became rough, as indicated by membrane dye FM4-64 (Fig. 3e). In contrast, when cells expressed cytosolic GFP alone or membrane-binding defective IniA mutants, the plasma membrane was rather smooth (Fig. 3f–h). These results suggest that IniA is targeted to the cell membrane by the LI loop in bacteria and can potentially remodel membranes.

**Nucleotide-independent association.** Nucleotide-dependent dimer formation via the GTPase domain is a common feature of dynamin-like GTPases. However, we discovered no such interface but a HB2-stacking dimer in the crystal packing of our structures. Symmetric interactions of a D333α7–R351α7' salt bridge and D337α7–H344α7', E402α8–H348α7' hydrogen bonds are observed between the antiparallel HB2s (Supplementary Fig. 3d). To assess dimer formation in solution, we performed analytical ultracentrifugation (AUC). Purified IniA behaved as a monomer, and remained so in the presence of GTP, GDP, GMPPNP, or GDP-AlF$_4^-$ (Fig. 4a). These results indicate that IniA does not undergo dimerization in solution.

To further test the homotypic interactions of IniA on the membrane, we performed a vesicle tethering assay. Purified IniA was incubated with liposomes and tethering was measured as increasing solution turbidity (absorbance at 405 nm, A$_{405}$; Fig. 4b). We found that IniA was reproducibly able to mediate membrane tethering in the absence of nucleotide (Supplementary Fig. 4d), likely because membrane-bound IniA molecules interacted *in trans*. Interestingly, tethering was reproducibly diminished when GTP was supplied to IniA (Fig. 4b; Supplementary Fig. 4d). In contrast, when the GTPase-defective mutant S50A/R63A was tested, *in trans* self-association was not affected (Supplementary Fig. 4f). The disruption of IniA-mediated tethering by $Mg^{2+}$ and the nucleotides is less likely due to charge alteration, because the addition of $Mg^{2+}$ and ATP did not affected tethering (Fig. 4b). To further confirm the tethering state, we labeled liposomes with rhodamine-PE, and visualized tethering by fluorescent microscopy. As expected, liposomes were evenly distributed in conditions where no elevation of A$_{405}$ was seen, and became clustered when IniA was added but nucleotide was omitted or when IniA, $Mg^{2+}$, and ATP were included (Fig. 4c). The tethering by IniA is reversible, since the A$_{405}$ dropped significantly when IniA was first added to establish tethering and GTP was then added (Fig. 4d). Finally, we tested whether the HB2 interface we saw in the structures plays a role in IniA homotypic interactions. When purified IniA triple-mutant D333K/D337K/E402K was incubated with liposomes, tethering of vesicles was not altered (Supplementary Fig. 4e). However, when the HB2 was deleted, tethering could no longer be observed (Supplementary Fig. 4f), suggesting a minor role of the HB2-stacking interface but a requirement of the HB2 in IniA self-assembly. Taken together, these results confirm that IniA forms a

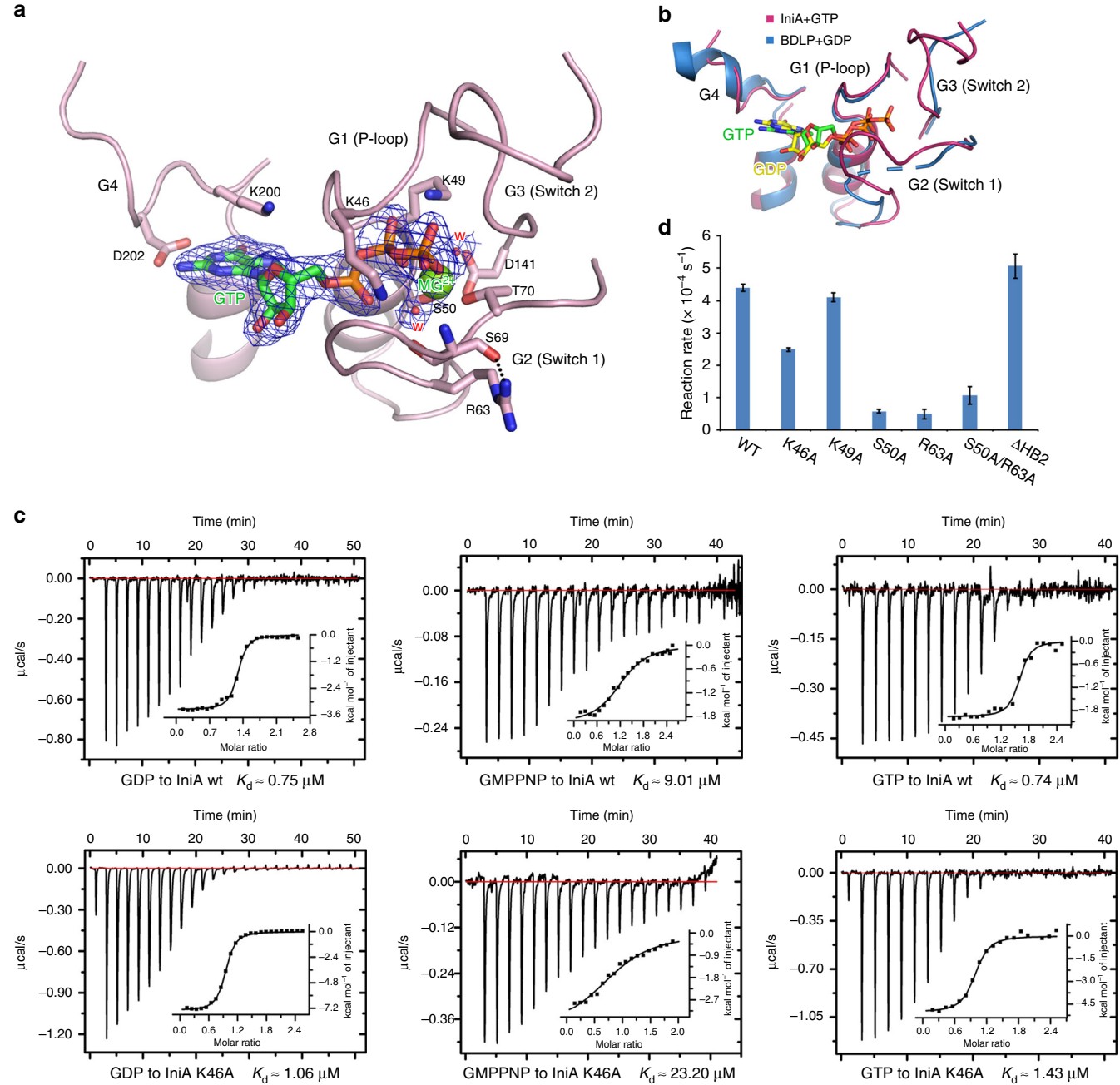

**Fig. 2** Analysis of the GTP-binding site. **a** The binding of GTP to IniA. GTP and nearby residues are shown as sticks. $Mg^{2+}$ and water molecules (W) are represented by spheres. The $2F_o-F_c$ electron density maps (contoured at 1.0 σ) of GTP, $Mg^{2+}$, and water molecules are shown as wire mesh (blue). The hydrogen bond between S69 and R63 is indicated by the dashed line. **b** Structural alignment of nucleotides and conserved motifs between IniA and cyanobacteria BDLP. The dashed loop represents un-modeled fragment in BDLP. **c** Binding affinity of GDP/GMPPNP/GTP for wild-type (wt) IniA and the K46A mutant measured by ITC. The dissociation constant, $K_d$, is given below. The data are representative of at least three repetitions. **d** GTPase activity of IniA and different mutants. The activities were measured by the reaction rate at 0.5 mM GTP. Each bar is the mean and SD of three measurements. The source data of Fig. 2d are provided in the Source Data file

nucleotide-independent association in the context of membranes, and GTP binding may cause conformational changes that interfere with nucleotide-independent homotypic interaction.

**Membrane remodeling by IniA.** Dynamin superfamily members are known to remodel membranes. Cyanobacterial BDLP has been shown to self-assemble and tubulate liposomes in the presence of GMPPNP[14]. When IniA was incubated with giant uni-lamellar vesicles (GUVs) containing PE-enriched lipids, it

interacted with the membrane and occasionally remodeled it into buds and clouds (Fig. 5a), reminiscent of the deformed plasma membrane in cells overexpressing IniA. Similar membrane puncta were also observed for the S50A/R63A mutant (Fig. 5a), suggesting that GTP hydrolysis is not necessary for puncta formation. As expected, no membrane association and deformation was seen with the LI loop-deleted mutant (Δ480–492, Fig. 5a). Interestingly, when GTP was mixed with IniA-decorated GUVs, rupture of GUVs was frequently observed, but no detectable GTP-dependent rupture occurred with S50A/R63A (Fig. 5b). We

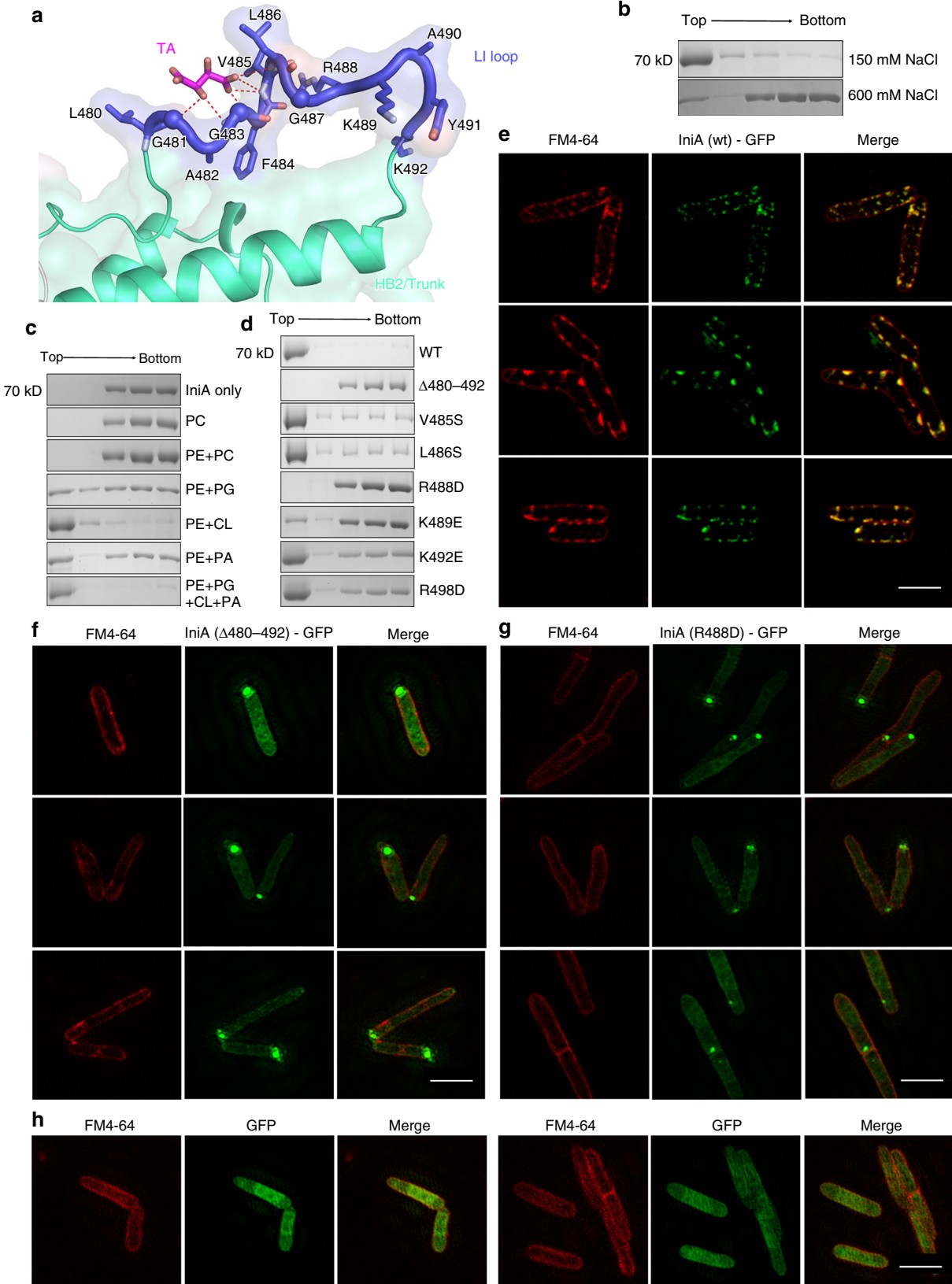

then tested whether GTP binding is sufficient for disrupting GUVs by IniA. A small portion of GUVs is broken when treated with wild-type IniA in the presence of GMPPNP. The number of GUVs remained the same after further incubation (Fig. 5b, c). In contrast, the addition of GTP caused continuous destruction of GUV by wild-type IniA. Approximately 80% GUV was ruptured after 45 min (Fig. 5d). No changes in GUV numbers were seen for either S50A/R63A or Δ480–492. These results indicate that IniA is capable of deforming and possibly cutting membranes, which requires its physical presence on the membrane. While

**Fig. 3** LI loop-mediated membrane association. **a** Residue conformation of the LI loop. Residues are shown as sticks, except that Gly is represented as a sphere. Tartaric acid (TA) interacts with the main chain N atoms in the LI loop (indicated by the dashed lines). **b** Liposome (*E. coli* polar lipids containing 78 mol% PE, 12 mol% PG, 6 mol% CL, 4 mol% PA) flotation assay showing the effect of NaCl concentration (150 mM and 600 mM) on IniA membrane association. **c** As shown in (**b**), but testing the effect of different lipids (100 mol% PC; 50 mol% PE, 50 mol% PC; 80 mol% PE, 20 mol% PG; 80 mol% PE, 20 mol% CL; 80 mol% PE, 20 mol%PA; 78 mol% PE, 12 mol% PG, 6 mol% CL, 4 mol% PA) for recruiting IniA. **d** As shown in (**b**), but with mutants at the LI loop. Δ480–492, L480-K492 was replaced by the GGGGSGGGGS linker. **e** IniA(wt)-GFP fusion protein was expressed in *M. smegmatis*, and their localization determined by GFP (green) and compared with that of FM4–64 (red) inserted in the membrane by confocal microscopy. Scale bars, 2 μm. **f–h** As shown in (**e**), the mutants IniA(Δ480–492)-GFP (**f**) and IniA(R488D)-GFP (**g**) and the control GFP (**h**) were also performed. Scale bars, 2 μm. The source data of Fig. 3b–d are provided in the Source Data file

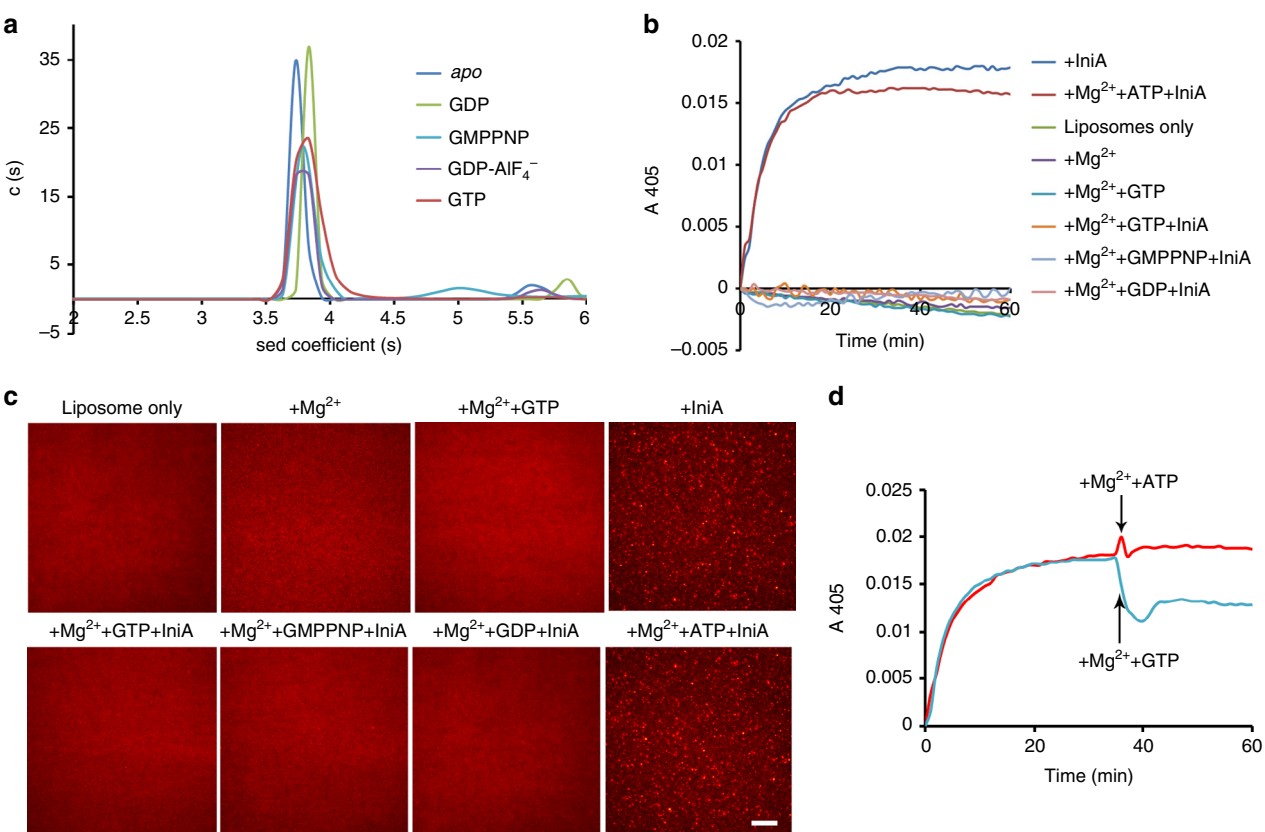

**Fig. 4** Nucleotide-independent oligomerization. **a** The molecular masses of IniA (theoretical molecular mass 65.9 kDa) were determined from sedimentation coefficients measured by analytical ultracentrifugation in the absence or presence of 2 mM GDP/GMPPNP/GTP or 2 mM GDP, 5 mM AlCl₃,10 mM NaF. The data are representative of at least three repetitions. **b** Liposomes (78 mol% PE, 12 mol% PG, 6 mol% CL, 4 mol% PA, final lipid concentration 2 mM) were mixed in various conditions, 2 μM IniA was added at initiation, and the absorbance was measured at 405 nm. The data are representative of at least three repetitions. **c** As shown in (**b**), the results were observed using a fluorescence microscope. Scale bar, 40 μm. **d** As shown in (**b**), 2 mM Mg$^{2+}$ and 5 mM GTP (or 5 mM ATP as a control) were added after a period of reaction. The source data of Figs. 4a, b and d are provided in the Source Data file

the deformation is nucleotide-independent, continuous cutting requires GTP hydrolysis.

Given the structural similarity between BDLP and MFN, BDLP has been speculated to mediate membrane fusion[22]. *Streptomyces* BDLP DynA has been reported to cause nucleotide-independent vesicle merging[8]. Therefore, we measured the fusion activity of IniA in a lipid-mixing assay. Purified IniA was mixed with two types of liposomes. The donor liposomes contained lipids labeled with nitrobenzoxadiazole (NBD) and rhodamine at quenching concentrations; fusion with the unlabeled acceptor vesicles led to fluorophore dilution and dequenching. We failed to detect IniA-mediated fusion, indicated by NBD fluorescence, with or without the addition of GTP (Supplementary Fig. 5a). These results suggest that IniA is less likely a membrane fusogen.

To this end, we tested whether IniA mediates fission similar to dynamin. Supported membrane tubes (SMrTs) were generated upon hydration of dry lipids with flow-induced extrusion on a glass surface. Fluorescently labeled IniA was subsequently passed through these lipid tubes in the fluidic chamber. In the absence of GTP, IniA efficiently coated SMrTs, but cleavage was undetectable (Fig. 6a). When GTP was added, fission was clearly noted (Fig. 6a; Supplementary Fig. 5b; see Fig. 6b for a series of sequential images and Supplementary Movie 1). Consistently, IniA formed puncta at fission sites. No fission was observed when GDP or GMPPNP was supplied (Supplementary Fig. 5c). We also tested GTPase-defective IniA double-mutant S50A/R63A, and captured no membrane cutting events. Similarly, no fission was observed when IniA R488D, a membrane binding-defective

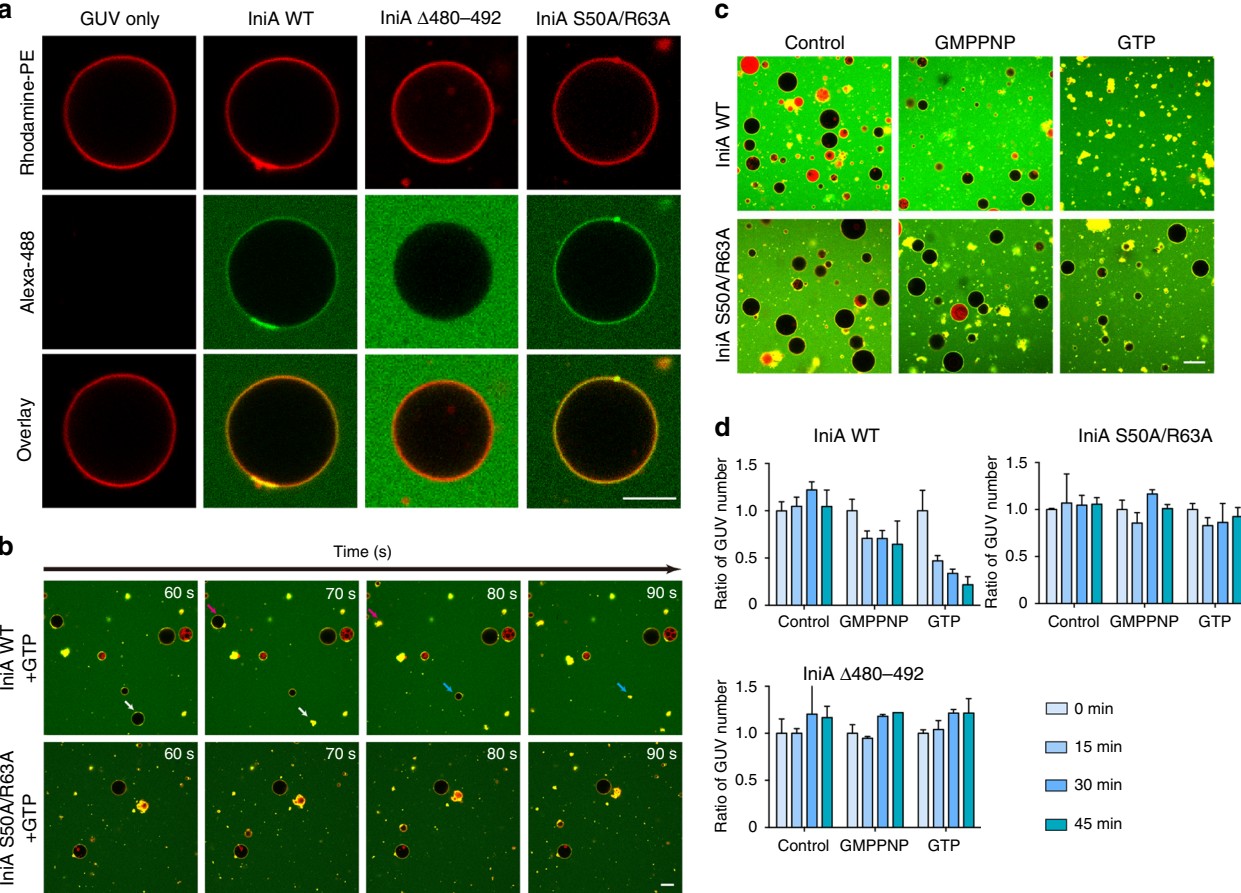

**Fig. 5** GUV rupture by IniA. **a** In vitro GUVs (red) binding assays show different binding ability of 15 μM WT IniA, Δ480–492, and S50A/R63A mutants (green). Scale bar, 10 μm. **b** As shown in (**a**), WT IniA and S50A/R63A mutant with the time course after 0.5 mM GTP/GMPPNP incubation for 10 min, the arrows indicate the rupture sites. Scale bar, 10 μm. **c** GUVs treated with 15 μM WT IniA and S50A/R63A mutant in the presence of 0.5 mM GTP/GMPPNP during 45 min, untreated GUVs as a control. Scale bar, 10 μm. **d** The diagram shows the statistical results in (**c**), the number of GUVs seen at indicated time points was counted and shown as relative values to that at the start point. Each bar is the mean and SD of three measurements. The source data of Fig. 5d is provided in the Source Data file

mutant, was used (Fig. 6a; Supplementary Fig. 5b), in this case, the mutant was not even retained on the membrane. Finally, when ΔHB2 was used, it attached to membrane tubules, but no fission was seen (Supplementary Fig. 5d). Taken together, these results indicate that IniA is capable of mediating membrane fission. The fission reaction appears to depend on GTPase activity, lipid interactions, and self-assembly of IniA.

Given that overexpression of IniA caused wrinkles of the plasma membrane, we suspected that IniA acts as dynamin-1 by tubulating and cutting cell membrane. Indeed, when 3D reconstruction was performed using SIM images of IniA-overexpressing *M. smegmatis*, membrane invaginations and intracellular vesicles could be seen (Supplementary Fig. 6). We then tested whether the membrane remodeling activity of IniA is involved in drug resistance. When *M. smegmatis* was treated with isoniazid, growth was severely retarded (Fig. 7a). Cells lacking IniA only exhibited marginally increased sensitivity when compared with wild-type, suggesting that endogenous IniA plays a minor role in isoniazid resistance of *M. smegmatis* (Fig. 7a). However, when wild-type IniA was overexpressed in IniA-deleted cells, some drug resistance was gained (Fig. 7b). Next, we tested whether IniA mutants convey similar resistance. GTPase-defective S50A/R63A mutant failed to increase growth in the presence of isoniazid (Fig. 7c). HB2 was also important, as overexpression of ΔHB2 caused very little drug resistance (Fig. 7c). The Δ480–492

mutant was partly effective in antagonizing isoniazid, whereas R488A mutant was more effective than the Δ480–492 mutant, but less potent than the wild-type (Fig. 7c). These results confirm that membrane remodeling related activities of IniA, including GTPase, self-assembly, and lipid interactions, play roles in IniA-mediated drug resistance.

## Discussion

Our structural and biochemical analyses revealed important features of IniA. Some characteristics are reminiscent of previously identified BDLPs. In summary, IniA has very slow GTP hydrolysis rates compared with other DLPs, exhibits a GTPase domain and HB combination, and binds to lipid bilayers and deforms membranes. Interestingly, IniA is unique in that it barely forms nucleotide-dependent dimers in solution. It likely forms nucleotide-independent homotypic interactions on membranes. The assembly of IniA unlikely depends on antiparallel HB2 stacking, but may resemble that of other BDLPs. The cyanobacteria BDLP form back-to-back association between the straightened HBs, including the Neck and the Trunk[14]. *Campylobacter jejuni* BDLPs have the same homotypic interactions. In addition, they possess an N-terminal extension that mediates heterotypic assembly between two isoforms from the same species[23]. These interfaces are all nucleotide independent.

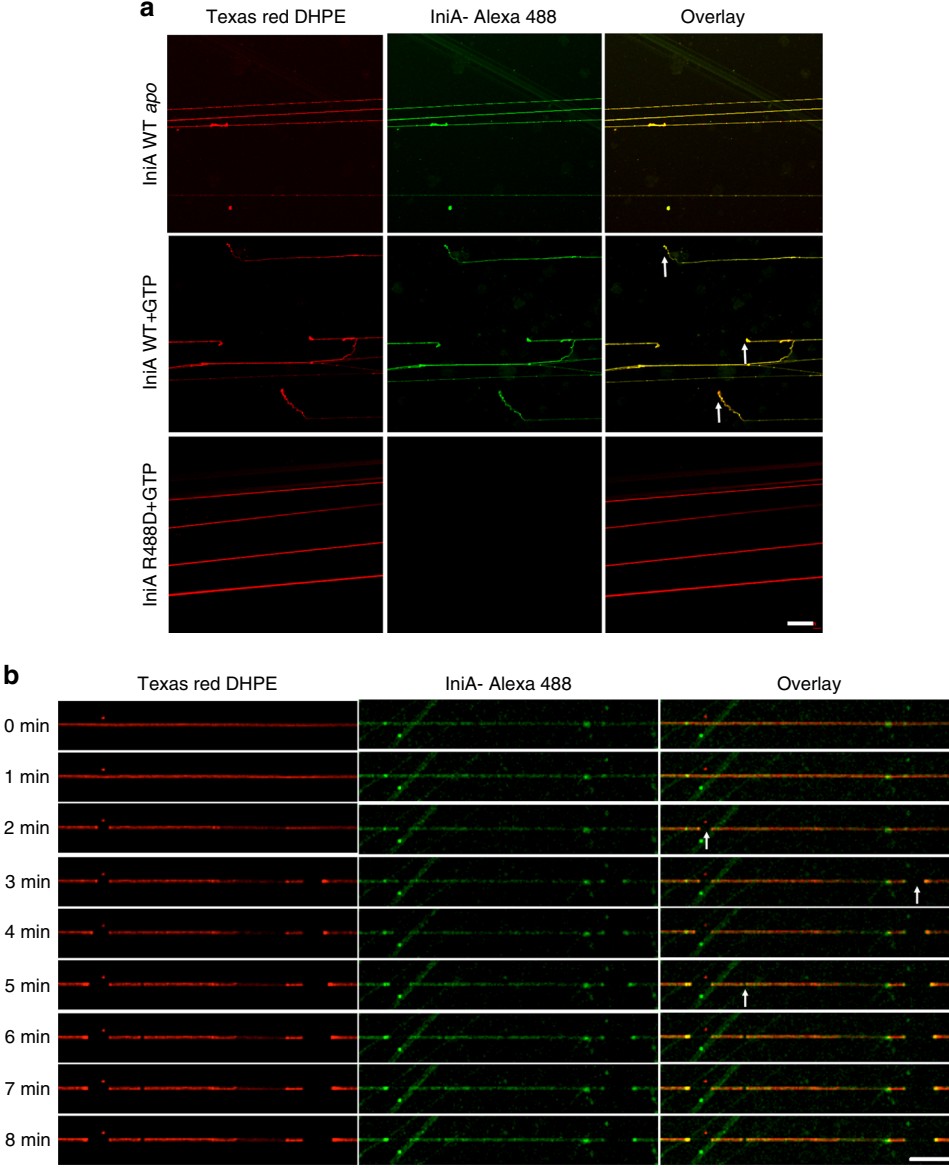

**Fig. 6** Membrane fission by IniA. **a** SMrTs (red) were treated with wild-type IniA (IniA WT) and R488D mutant in the absence or presence of 5 mM GTP. Protein localization and tube cleavage were monitored using confocal microscopy. The arrows indicate the cleavage sites. Scale bar, 20 μm. **b** As shown in (**a**), but with wild-type IniA recorded along with the time course after GTP addition. The arrows indicate the cleavage sites. Scale bar, 20 μm

The potential LI loop in LeoA and B-inserts in other DLPs often point directly toward the membrane at the tip of the stalk[11], whereas that of IniA is parallel to HB2. Given the orientation of the LI loop, IniA likely lies on membranes with some curvature. Consistently, when IniA was placed on supported lipid bilayer, it forms almost flat hexamers[7]. The straightening of the stalk, which was not observed here, is presumably linked to conformational changes in the GTPase domain. Motions in the GTPase core may trigger inward swing of the HB1/Neck, and at the same time, release the α9-loop and its associating HB2/Trunk. They may reorient the LI loop that connects to the α9-loop, even though its membrane association affinity would not change much as indicated by our flotation experiments. It is surprising that IniA maintains its conformation after GTP binding in our structures. The bent conformation of the stalk we observed in the GTP-bound state is possibly physiologically relevant before membrane binding. A different conformation of GTP-bound state in the membrane could be expected, which, however, cannot be crystallized in the absence of membranes. The actual conformation

and organization of IniA during membrane remodeling requires further analysis. In any case, the GTP-hydrolysis cycle-dependent conformational changes are most likely responsible for rearrangement of the potential helical filament of IniA.

The detected scission activity by IniA is relatively low. It is possible that help from its homolog IniC is critical. Evidence of cooperative BDLP activity comes from the *Cj*DLP1/2 tetramer structure[23]. The GTPase activity and lipid association is induced by heterotypic assembly of the two homologous BDLPs. We were unable to analyze IniC structurally or biochemically, due to its poor stability after purification. IniC may regulate IniA activity and/or form hetero-oligomers with IniA. IniA does not have an N-terminal extension as seen in *Cj*DLPs. However, IniC may have such an extension that leads to heterotypic interactions.

Secondary structure alignment suggests that the molecular architecture of IniA is close to MFN[24], in particular with the long loop after α8 (Supplementary Fig. 1). The assembly of the IniA Trunk predicts how the MFN HB2 would look like. The complementation of the N-terminal IniA HBs by C-terminal α11a

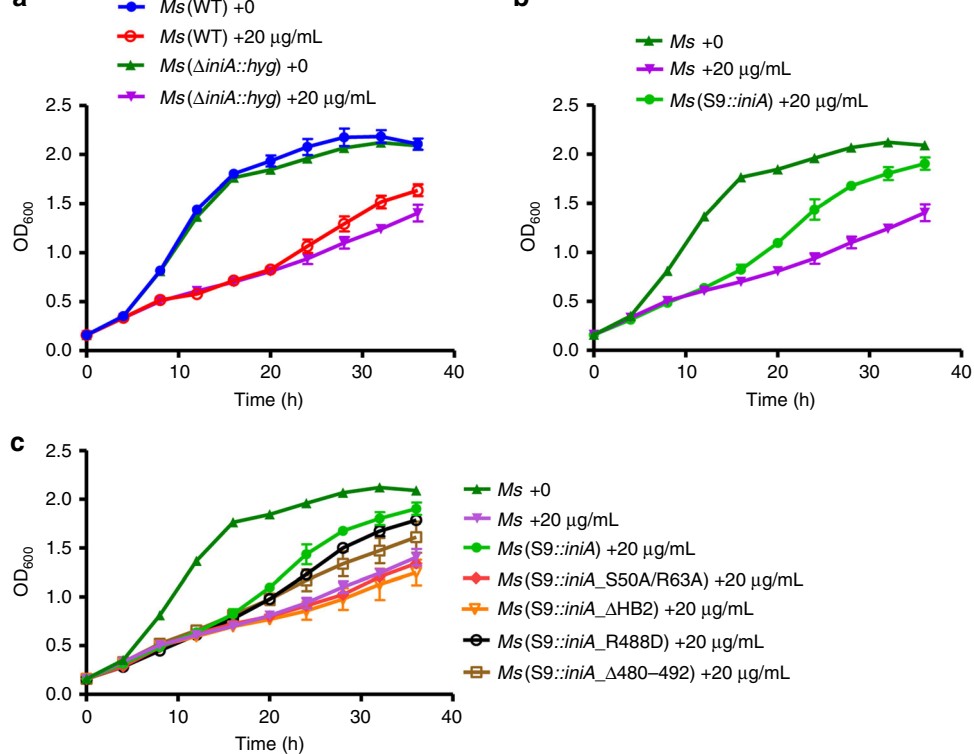

**Fig. 7** IniA-mediated isoniazid resistance in *M. smegmatis*. **a** Growth curves of wild-type *M. smegmatis* [*Ms* (WT)] and *iniA*-deleted strain [*Ms* (Δ*iniA::hyg*)] in the presence or absence of 20 μg mL$^{-1}$ isoniazid. **b** Growth curves of *Ms* (Δ*iniA::hyg*) with overexpression of wild-type IniA (S9::*iniA*) in 20 μg mL$^{-1}$ isoniazid, compared with no antibiotics or no IniA overexpression. **c** As shown in (**b**), overexpression of mutant IniA, including S50A/R63A, ΔHB2, R488D, and Δ480–492. Each bar is the mean and SD of three measurements. The source data of Fig. 7a–c are provided in the Source Data file

and α11b in the full-length protein suggest that the C-terminus of MFN would similarly remain in the cytosol to complete the folding of MFN and not likely to project into the inter-membrane space as proposed recently[25]. Despite the structural similarity with MFN, we did not detect fusion activity for IniA. Instead, we found that IniA mediates fission similar to dynamins. A tandem BDLP, DynA, has been reported to mix lipids in a nucleotide-independent manner[8]. However, increasing concentrations of Mg$^{2+}$ ion, which neutralizes negative charges on the liposomes, caused vesicle aggregation and subsequent nonspecific lipid mixing. Purified *Cj*DLPs, with certain combinations, can tether vesicles in vitro, but no nucleotide is needed, and no subsequent fusion is detected[23]. These findings are consistent with what we have observed with IniA, which could represent the ability of these BDLPs to form membrane-induced nucleotide-independent self-assembly.

It is rather intriguing that two structurally similar proteins, IniA and MFN, play opposite roles in membrane remodeling. Similar to dynamins, IniA undergoes nucleotide-independent self-assembly, which requires at least the HB2/Trunk and would be important for wrapping around a constricted membrane area. However, nucleotide-dependent dimerization of GTPase domains is not detected in solution and may occur transiently when assembling *in cis* on membranes. GTP binding causes conformational changes, as indicated by our tethering assay; GTP hydrolysis is needed for fission, but likely induces conformational changes that would change any helical lattice. Conversely, MFN forms strong nucleotide-dependent dimerization between two GTPase domains. MFN also forms nucleotide-independent clustering, but like ER-fusing dynamin ATL[26], it is through the TM segments. The nucleotide-dependent dimerization in trans is thus critical for bringing two membranes together for subsequent fusion.

Our results indicate that the fission activity of IniA is involved in mycobacterial drug resistance. Given that INH and EMB block cell wall biogenesis, IniA probably contributes to the maintenance of the plasma membrane integrity. For example, fission of the compromised areas could be used for cell membrane repair[27]. Consistent with such prediction, overexpressed IniA in Mycobacteria clearly remodels the plasma membrane.

## Methods

**Constructs and strains.** The DNA sequence encoding full-length IniA (residues 1–602) from *M. smegmatis* mc$^2$155 was PCR-amplified using Phanta Max DNA polymerase (Vazyme) and inserted into the NcoI-HindIII sites of pET-28a vector (GE Healthcare) using ClonExpress®II (One Step Cloning Kit, Vazyme). This construct includes a C-terminal 6 × His-tag. For expression in *M. smegmatis*, the encoding sequence of IniA-GFP fusion protein was inserted into the SspI site of the engineered plasmid pMV261 and the mutant construct (residues L480-K492 was mutated into GGGGSGGGGS liker) was generated by recombinant circle PCR. Site-directed mutagenesis was performed using the TaKaRa MutanBEST Kit. The mutants were introduced by the PCR method using the IniA expression plasmid as a template, with pairs of primers encoding the mutations at the sites of substitution. DNA sequencing of the constructs was performed to validate that the mutagenesis experiments were successful. All primer sequences used in this study are shown in the Supplementary Table 1.

*E. coli* BL21 (DE3) ATCC strain (Supplementary Table 2) was used to express IniA for purification. The *M. smegmatis* mc$^2$155 strain was used to observe cellular localization of IniA.

**Protein expression and purification.** The IniA constructs were transformed into *E. coli* BL21 (DE3) for protein overexpression. The bacteria were cultured in Luria-Bertani media at 37 °C to an OD$_{600}$ of 0.6. Protein expression was induced by the addition of 0.2 mM IPTG for 20 h at 16 °C. Cells were harvested after centrifugation at 4000 *g* for 30 min, resuspended in Buffer A (20 mM Tris, pH 8.0, 150 mM NaCl), lysed under ultra-high pressure, and centrifuged at 39,000 *g* for 30 min to remove cell debris. The supernatant was then collected. The soluble protein was isolated by Ni-NTA chromatography (GE Healthcare) and further purified by gel filtration chromatography (Superdex-200 10/300GL increase; GE Healthcare) in

Buffer A. Fractions containing IniA were pooled, concentrated, flash-frozen in liquid nitrogen, and stored at −80 °C for later use.

**Crystallization and structure determination**. Crystal-screening experiments were performed at 16 °C using the sitting-drop vapor-diffusion method and a Gryphon-LCP robot (Bioray). Each drop consisted of 200 nL of protein solution and 200 nL of reservoir solution from commercial crystal-screening kits. To obtain crystals of the complex, the protein was concentrated to 13 mg mL$^{-1}$ in the presence of 5 mM GTP and 5 mM magnesium chloride. The reservoir solution consisted of 0.1 M ammonium tartrate dibasic (pH 7.0) and 12% (w/v) PEG3350. Platinum derivatives were prepared by soaking crystals in the reservoir solution supplemented with 5 mM K$_2$Pt(NO$_2$)$_4$ for 4 h prior to cryoprotection. To obtain IniA crystals in the absence of GTP, 10 mg mL$^{-1}$ protein and reservoir solution containing 0.05 M citric acid, 0.05 M BIS-TRIS propane (pH 5.0), and 16% (w/v) PEG3350 was used. Crystals were then flash-cooled in liquid nitrogen with 20–30% (v/v) methanol as the cryoprotectant. Diffraction data were collected on beamline BL18U1 and BL19U1 of the Shanghai Synchrotron Radiation Facility (SSRF) as well as BL41XU of Spring-8. HKL2000[28] was used for data integration, scaling, and merging. The initial phasing problem was solved by single-wavelength anomalous dispersion (SAD) using PHENIX[29] for the GTP-bound IniA. The model was initially built in PHENIX and manually adjusted in COOT[30]. Refinement was carried out using PHENIX allowing the coordinates to vary and in real space. Each atom was refined to have an individual isotropic B-factor. The rigid body motion of the protein was fitted according to TLS parameters. The nucleotide-free structure was solved by molecular replacement with the program PHASER[31] using the GTP-bound structure as a search template. Data collection and refinement statistics are provided in Table 1. Structural figures were prepared using PyMol 2.1.

**GTPase activity assay**. GTPase assays were performed using the Enzchek phosphate assay kit (Invitrogen). Reactions were performed in a 100 μL volume with 20 μL 5× reaction buffer, 200 μM 2-amino-6-mercapto-7-methylpurine riboside (MESG), 0.1 U purine nucleoside phosphorylase (PNP), and 10 μM wild-type or mutant IniA protein. The proteins were incubated for 25 min at 37 °C in a 96-well plate (Corning). Reactions were initiated by the addition of 0.5 mM GTP (Thermo Fisher Scientific). The absorbance was measured at 360 nm every 30 s over 30 min at 37 °C using a Tecan infinite M200 PRO reader. The reaction rate was calculated based on a standard curve.

**Isothermal titration calorimetry**. ITC was performed at 20 °C with a MicroCal iTC200 instrument (GE Healthcare). Proteins were prepared in ITC buffer containing 20 mM HEPES (pH 7.5), 150 mM NaCl, and 4 mM MgCl$_2$. GDP, GMPPNP, and GTP, all at a final concentration of 2 mM, were directly dissolved in ITC buffer. The concentration of IniA protein (or the mutant K46A) for GDP and GMPPNP/GTP was 100 μM and 80 μM, respectively. The acquired ITC data were analyzed by the Origin 7.0 (GE Healthcare) program using the One Set of Binding Sites fitting model.

**Flotation assay**. Lipids (76.5:12:6:4:1.5 mole percent POPE:*E. coli* PG:*E. coli* CL: POPA:rhodamine-DPPE and other lipid composition as indicated) were dried to a film, hydrated with 20 mM HEPES (pH 7.4) and 150 mM NaCl (or 600 mM NaCl), and extruded through polycarbonate filters with a pore size of 100 nm. Liposomes (final lipid concentration 2.5 mM) were mixed with 2 μM wild-type with or without various nucleotides, or mutant IniA and incubated at 37 °C for 20 min. The 30 μL mixture of proteins and liposomes was mixed with 100 μL of 1.9 M sucrose and overlaid with 100 μL of 1.25 M sucrose and 20 μL of 0.25 M sucrose, all in 25 mM HEPES (pH 7.4). The samples were centrifuged in a Beckman TLS 55 at 174,000 g at 4 °C for 80 min. The gradient was fractionated into five 50-μL fractions, and the samples were analyzed by SDS-PAGE.

**Cellular localization assay**. The pMV261 constructs were transformed into wild-type *M. smegmatis* mc$^2$155 (Supplementary Table 2), and transformants were selected using 20 μg mL$^{-1}$ carbenicillin and 10 μg mL$^{-1}$ kanamycin as the antibiotics. Luria-Bertani media was used to culture the bacteria at 37 °C to an OD$_{600}$ of 0.6–0.8, and expression was induced using 0.2% (w/v) acetamide at 16 °C for 24 h. Cells were harvested and resuspended in PBS buffer and washed three times. A 10 μL aliquot of these cells was stained with 20 μg mL$^{-1}$ FM4-64 (diluted with PBS using 2 mg mL$^{-1}$ FM4-64 in DMSO) for 1 min to label the bacterial membrane[32]. To obtain optimal images, immersion oils with refractive indices of 1.512 were used for bacterial cells on glass coverslips. 3D-SIM images were acquired on the DeltaVision OMX V3 imaging system (GE Healthcare) with a × 100/1.40 NA oil objective (Olympus UPlanSApo), solid-state multimode lasers (488 nm, 561 nm), and electron-multiplying CCD (charge-coupled device) cameras (Evolve 512 × 512, Photometrics). Serial Z-stack sectioning was done at 125 nm intervals for SIM mode. Three-dimensional reconstructions were performed using IMARIS 8 software (Bitplane AG, Switzerland). IniA and cell membrane surfaces were created using the Surfaces tool with automatic settings based on the fluorescent signals from GFP and FM4-64.

**Analytical ultracentrifugation**. Purified IniA (32 μM) was used for AUC in a buffer containing 20 mM HEPES (pH 7.5), 150 mM NaCl, and 4 mM MgCl$_2$. Sedimentation velocity experiments were performed at 10 °C in a proteome Lab XL-1 Protein Characterization System (Beckman Coulter). Before centrifugation, the nucleotide (2 mM GDP; 2 mM GMPPNP or 2 mM GDP, 5 mM AlCl$_3$, 10 mM NaF) was added to 400 μM protein. All interference data were collected at 140,000 g using an An-60 Ti rotor (Beckman Coulter). The AUC data were processed according to a c(M) distribution model.

**Membrane tethering and lipids mixing assay**. To prepare liposomes, lipids were dried to a film, resuspended in 20 mM HEPES, pH 7.5. Liposomes were prepared from this mixture by extrusion through polycarbonate membranes (Avanti Polar Lipids) with a pore size of 100 nm using an Avanti Mini-Extruder (Avanti Polar Lipids) followed by ten freeze-thaw cycles. Donor liposomes (75:12:6:4:1.5:1.5 mole percent POPE:*E. coli* PG:*E. coli* CL:POPA: rhodamine-DPPE:NBD-DPPE, final lipid concentration 0.5 mM) and acceptor liposomes (78:12:6:4 mole percent POPE:*E. coli* PG:*E. coli* CL:POPA, final lipid concentration 1.5 mM) were generated. For the membrane tethering assay, liposomes were mixed with or without 2 μM IniA in various conditions (see Fig. 4; Supplementary Fig. 4), and A$_{405}$ was measured at 37 °C. For the lipids mixing assay, liposomes were pre-mixed with or without nucleotides and Mg$^{2+}$, and the fluorescence intensity of NBD was monitored with an excitation of 460 nm and emission of 538 nm after 2 μM IniA was added or not. The initial NBD fluorescence was set to zero, and the maximum fluorescence was determined after the addition of 10 μL 2% (w/v) dodecylmaltoside (DDM).

**Preparation of GUVs and the rupture assay**. Giant unilamellar vesicles (GUVs) lipids contain 40 mol% PG, 39 mol% POPE, 10 mol% CA, 8 mol% PI, 2 mol% DOPS, and 1 mol% rhodamine-PE (Avanti). GUVs were made by electroformation. Briefly, lipid mixture in chloroform was deposited on indium-titan oxide glass slides and dried for 60 min in a vacuum to evaporate all solvents. GUVs were electroformed in 2 mL of 200 mM sucrose by a Pulse Generator for 6 h at room temperature. In all, 15 μM of purified IniA WT, IniA Δ480–492 and IniA R50A/S63A, with a ybbr tag at the C-terminus were labeled by Alexa Flour 488-CoA, were mixed with GUVs and incubated at room temperature for 15 min. Next, the GMPPNP and GTP were added into GUVs-protein system for 0.5 mM and incubated for 15 min.

GUVs were treated with IniA WT, IniAΔ480–492, IniA R50A/S63A in 0.5 mM GTP/GMPPNP during 45 min, and untreated GUV was used as the control. For every group, five random field photographs were taken by confocal microscopy (Zeiss LSM 880), and counts the number of GUVs with a diameter greater than 10 μm. When GTP/GMPPNP was added, this is recorded as time = 0, and the number of GUVs in Time 0 records as 1.0.

**SMrT-based membrane fission assay**. The SMrT-based membrane fission assays were performed with modifications[33,34]. Briefly, lipids (77:12:6:4:1 mole percent POPE:*E. coli* PG:*E. coli* CL:POPA:Texas red DHPE) were diluted to a final concentration of 1 mM total lipid in chloroform. A small aliquot (∼1 μL) was spread on a freshly cleaned PEG8000-coated glass coverslip and kept under high vacuum for 5 min to remove traces of chloroform. A flow cell (FCS2 system, Bioptechs) was assembled by placing a 0.1-mm silicone spacer between the PEGylated coverslip and an ITO-coated glass slide. The flow cell was filled with 20 mM HEPES (pH 7.4) and 150 mM NaCl and left undisturbed for 10 min at room temperature. SMrTs were created by extrusion of the large vesicles formed during hydration to narrow membrane tubes by flowing excess buffer at high flow rates. IniA with a ybbr tag at the C-terminus was labeled by Alexa Fluor 488-CoA[35]. In brief, 0.1 μM Sfp, 5 μM biotin-CoA, and 5 μM ybbR-tagged protein were mixed into a total volume of 100 μL containing 10 mM MgCl$_2$ and 50 mM HEPES pH 7.5. The reaction mixture was incubated at room temperature for 30 min. Ybbr-tagged proteins in cell lysates can be directly labeled by Sfp. Then, 500 μL of 30 μM labeled IniA was introduced into the chamber at a low flow rate. Next, 2 mL 20 mM HEPES (pH 7.4) and 150 mM NaCl was pumped into the chamber to wash away unbound proteins. Subsequently, 500 μL of 20 mM HEPES (pH 7.4) and 150 mM NaCl supplemented with 5 mM MgCl$_2$ and 5 mM GTP/GDP/GMPPNP was pumped into the chamber to initiate the fission of SMrTs. All reactions were carried out at 37 °C controlled by a μ-environmental controller (FCS2 system, Bioptechs) and imaged by confocal microscopy (LSM710 META, Zeiss).

**Drug-resistance assay**. *M. smegmatis* strains were grown in 7H9 media supplemented with 10% (v/v) ADS (50 g BSA, 20 g glucose and 9 g NaCl dissolved in 1000 mL ddH$_2$O), 0.5% (v/v) glycerol, 10 μg mL$^{-1}$ kanamycin and 0.1% (v/v) Tween-80. Cells were cultured at 37 °C to an OD$_{600}$ of 0.6–0.8, and diluted to OD$_{600}$ of 0.1. Indicated amounts of isoniazid were then added, and OD$_{600}$ was measured every 4 h, using 96 ½ area plate (Corning) in Multimode Plate Reader (EnSpire). Data analyses were performed using GraphPad Prism 5.0.

**Reporting summary**. Further information on research design is available in the Nature Research Reporting Summary linked to this article.

## Data availability

Data supporting the findings of this manuscript are available from the corresponding authors upon reasonable request. Coordinates and structure factors have been deposited in the Protein Data Bank under accession numbers 6J72 and 6J73. The source data underlying Figs. 2d, 3b–d, 4a, b, d, 5d and 7a–c, and Supplementary Figs. 4a–g and 5a, b are provided as a Source Data file.

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

## Acknowledgements

We thank the staff members of the Protein Expression and Purification System, Integrated Laser Microscopy System and Electron Microscopy System at the National Facility for Protein Science in Shanghai (NFPS), Zhangjiang Lab, China for providing technical support and assistance in data collection. We thank Thomas Pucadyil, Xiaochen Wang, and Yubing Liu for helps with SMrT device. We would like to thank Shuoguo Li and Yun Feng from Center for Biological Imaging (CBI), Institute of Biophysics, Chinese Academy of Science for their help of taking and analyzing SIM images. We also thank Ying Zhang and Xiaoyun Pang for material and technical support. We are extremely grateful to staff members at beamlines BL18U1, BL19U1 and BL17U1 at Shanghai Synchrotron Radiation Facility (SSRF), as well as BL41XU at SPring-8 for their instrument support and technical assistance. This work was supported by grants from the Strategic Priority Research Program of the Chinese Academy of Sciences (Grant no. XDB08020200 to Z. R.), the National Key Research and Development Program (Grant nos. 2016YFA0500201 to J.H. and 2017YFC0840300 to Z.R.), the State Key Development Program for Basic Research of the Ministry of Science and Technology of China (973 Project Grant Nos. 2014CB542800 and 2014CBA02003 to Z.R.), National Natural Science Foundation (Grant Nos. 813300237, 81520108019 to Z.R, 31630020 to J.H., and 31500607 to J.L.), and the Strategic Priority Research Program (Pilot study) Biological basis of aging and therapeutic strategies of the Chinese Academy of Sciences (grant XDPB10).

## Author contributions

Z.R. initiated and supervised the project. J.L., J.H., and M.W. contributed to the overall study design. M.W. made all constructs, purified proteins, and obtained crystals. M.W. and J.L. collected diffraction data and solved structures. M.W. and X.G. performed functional experiments with help from B.Z., J.R., F.C., Y.R., A.L., B.Y., and X.Y. The data were analyzed by M.W., X.G., B.Z., F.C., Y.R., A.L., X.Y., L.W.G., J.L., J.H., and Z.R. The paper was written primarily by J.H., M.W., and J.L. with contributions from the other authors.

## Additional information

**Competing interests:** The authors declare no competing interests.

