## [Peer Review File · Nature Communications]

Reviewers' Comments:

Reviewer #1:

Remarks to the Author:

Mycobacterial dynamin-like protein IniA mediates membrane fission: Wang et al. 2018

Wang et al. show the crystal structure of *Mycobacterium smegmatis* IniA in both the apo and GTP-bound state. IniA is a dynamin homologue and has been shown to be upregulated in the presence of key tuberculosis drug treatments and to contribute to antibiotic resistance. It is therefore an important system to work on and it is really exciting to see the crystal structure. The structure has implication both for tuberculosis drug treatment and for the dynamin field in general, which still is yet to fully clarify mechanisms of membrane fission and fusion. In principle the structure with its coupling to membrane fission is of sufficient conceptual advance for publication in Nature Comms. However, whilst Wang et al. are to be applauded for the many techniques they have used to dissect the in vitro mechanism of IniA, the manuscript is weakened by a lack of support for some of the key conjectures. Many of the obvious questions that their experiments uncover are not followed up sufficiently, or at least are not shown here. The manuscript therefore is quite limited in places for this level. Also, the discussion is weak and there are some apparent misconceptions of some dynamin biology although this may be convoluted with a language issue.

Major:

L32: The authors state that IniA associates homotypically via HB2. Whilst I don't doubt that IniA will self-assemble via HB2 in some way, there is little to support this statement in the manuscript. The only evidence is the in vivo D333K/D337K/E402K mutant which shows a partial effect. Does this mutant inhibit tubulation or tethering in vitro for example as well? This experiment would greatly support the authors hypothesis. Also, it should be clarified that if the D333K/D337K/E402K interface is involved in self-assembly it may well be in a configuration totally different to that observed in the crystal structure with the anti-parallel HB2 association. Without some sort of supporting data the crystal contact in itself is not sufficient evidence for an assembly interface.

L34-36: IniA resembles mitofusin in that IniA closely resembles BDLP whose overall fold resembles that of Mitofusin 1. The actual conformation of IniA observed (hinge 1 closed) (is that what is meant by 'configuration'?) does not closely resemble Mitofusin 1 which has not yet been shown to hinge 1 close.

L36-38: This sentence is quite unclear and appears to be a general statement on dynamin biology. It doesn't have much to do with the data in the manuscript for example, IniA does not show G-dimerisation. This very much weakens the abstract and the impact of the paper.

L97: Crystal structure of IniA have been obtained in both a GTP-bound and apo state. The RMSD between them is just 0.5 Å. The lack of any conformational change outside of the nucleotide binding pocket is surprising although not necessarily a concern. In some dynamin systems, the crystallization process has ultimately determined conformation irrespective of nucleotide state (Byrnes & Sondermann PNAS 2011 and Wenger et al. Plos One 2013). The authors should comment on the total lack of conformational change observed between crystal states outside of the nucleotide binding pocket, and whether they believe the GTP state observed is physiologically relevant or whether crystallization may dominate the conformation here.

Also, does IniA crystallise in the presence of GDP? Presumably if there is no conformational change in solution between apo and GTP states then the GDP state is equivalent and crystallises? Similarly, does IniA crystallise in the presence of GTP non-hydrolysable analogues such as GMPPNP or GTP-γ-S? Presumably this gives the same crystal form as the GTP bound? It is not being asked that the authors check all these nucleotide states if they have not already done so. It is just trying to understand whether the GTP state observed with absolutely no conformational change is valid here.

L. 100 and Fig. 1. It would be helpful if new terminology for the neck and trunk domains could be avoided ie. HB1 and HB2. These are clearly very similar structural motifs as has previously been observed for BDLP and it would be helpful for the field if new nomenclature wasn't used for each new structure.

L170-178: Interesting IniA tethering activity is shown in the absence of nucleotide. The addition of GTP abolishes this tethering. This assay has the potential to give some exciting insight into IniA activity and it is frustrating that the study is so limited and lacking in many obvious variations that could easily have been done at the time. For example, 1) what happens if GTP is added after a time lag once tethering has already observed. Does turbidity then decline? 2) What happens with GDP? 3) What happens with GMPPNP? The study would be greatly powered if all these experiments were cross-checked with negative stain EM to visually validate what was going on and to help build up a compelling picture of IniA mediated membrane fission. For example, where tethering is observed in the absence of nucleotide as in Fig. 3C are protein-lipid tubes observed by EM? Are these then abolished when nucleotide is added or do they never form when GTP is included from the beginning?

L212-213 and Fig. 4E: Great to see apparent membrane localization of IniA in the cell and convincing that the GS-linker abolishes membrane localization. However, for the top row does WT mean +IniA-GFP? It would be good to see wild type cells with no IniA-GFP stained with FM4-64. This should look like GS-linker panels FM4-64 with localization in a nice red ring right? Therefore looking at the FM4-64 panel on the top row it is unclear to me why the membrane appears perturbed and punctate, and does not look like the other FM4-64 panels. Do the authors think that IniA-GFP is perturbing the membrane? Might this be contributing to the punctate localization pattern? Although not essential, does purified IniA-GFP tubulate liposomes, so is it functional? This should be discussed.

L227: This is really exciting to see the EM IniA protein-lipid tubes in the presence of GMPPNP. Again perhaps these experiments were done but considered not important to mention, but what happens when GTP is added, or GDP? Critically, does IniA tubulate in the absence of GTP? This is important as earlier the authors state that tethering occurs in the absence of GTP so we should see some effect right? Also what happens to IniA with GTP or GDP in the absence of lipid? Presumably just particles consistent with monomer is observed (ie no self assembly) given the AUC is monomer for apo or GMPPNP state at high concentration. It would be reassuring to know these conditions had all been checked though by EM. If the authors have access to cryo-EM showing the effect of GTP on liposomes, this would really help prove this idea that IniA induces membrane fission if vesiculation is observed etc..

L241-248: It is to the authors credit that they are using so many different techniques to probe the in vitro function of IniA and the use of SMrTs is compelling→→. However, similar to the turbidity assay and EM studies, again a little frustrating that obvious experiments such as doing a full nucleotide screen wasn't undertaken at the time and presented. For example, with the SMrT experiment, when GMPPNP is added presumably no cleavage is detected? What happens with GDP or the S50A hydrolysis deficient mutant? These are useful controls that would help power the final conclusion to the reader that IniA GTP turnover really induces membrane fission.

L260: This is rather weak as no dimer is described in solution. The crystal dimer may just be a crystal interface. Also, in BDLP it is both lipid and nucleotide binding required for stalk straightening (hinge 1 opening). This needs to be clarified by the authors how IniA polymerization is triggered. Does polymerisation occur just by lipid binding or is the addition of nucleotide required too?

L281-282: 'Usually tweaks the oligomerization pattern'. This is vague and should be clarified as

GTP hydrolysis is thought to induce large scale conformational changes that will significantly change the helical lattice and ultimately breaks up the filament during membrane fission.

L282-291: 'Touching between GTPase domains.. if it exists at all'. There is compelling data in the literature for the formation of the G-dimer in many dynamin systems (Dynamin1, DNMM1-L, BDLP, atlastins, GBP1 etc).

L289-291: 'Fission proteins generally emphasize... nucleotide-dependent association'. This is not very clearly written so is hard to understand but as I read it the dynamin literature just does not support this statement at all. Overall, the final paragraph is not supported by the dynamin field literature and needs reconsidering.

Overall, the discussion is quite limited, should be expanded and more detailed. For example, the possible nature of the protein-lipid tubes should be discussed- do the authors speculate formation of the G-dimer and BDLP-like trunk self-association or Dynamin 1-like interface 2 stalk self-association (this is closer to what is observed in the crystal packing). Or how does the low resolution IniA EM hexamer observed in Colangell et al. Mol Micro 2005 fit with their crystal structure? Also, many bacterial dynamins function as dimers (DynA in Bacillus and DynA/B in Streptomyces). It would be good if IniC in the IniA operon could be acknowledged and its potential role in activating and co-functioning with IniC discussed briefly.

Minor:

L56. Efflux pump role highly speculative

L62-68. As stated earlier, better to use BDLP terminology for neck and trunk rather than introducing new HB terminology. Writing is unclear also. It is the neck domain that connects the GTPase domain to the trunk. Dimerization is significantly mediated by the G-dimer interface as well as the paddle in BDLP. Please can this be clarified.

L71. Spirals is technically incorrect terminology. These are helices as the radius does not decrease with each turn.

L74-75. The LeoA neck does not readily superimpose on the BDLP neck. It has a slightly different conformation so the neck comparison should be avoided.

L84. Can be applied is a little strong here as this is still speculative.

L85-90. No role in cytokinesis has been suggested, only a speculative role in determining thylakoid morphology and cell shape. Also, it is unclear whether BDLP mediates fission or fusion reactions. So far a role in membrane fusion has speculatively been assigned due to the low nucleotide turnover rate in comparison to fission dynamins and absence of assembly stimulated turnover. At this point, suggesting a fission phenotype should be avoided. Opa1 for example generates well-ordered protein-lipid tubes and has a fusion phenotype so there is no rule of thumb that protein-lipid tube formation = fission dynamin.

L116-117. The sentence is confusing- IniA does not form an HB open conformation. Are the authors describing BDLP here?

L141. BDLP-GMPPNP has only been determined at low resolution by cryo-EM so the position of K46 should not be commented upon here.

L143. Please label the G2 loop on Fig. 2A. Does the U shaped flap on top of the GTP mean that the G-dimer would be inhibited from forming? The authors should clarify this as it may be important for IniA function if this flap precludes G-dimerisation.

L168-169: The anti-parallel HB2 association observed in the crystal is not supported by the AUC. 'Rather weak' in solution suggests that some dimerization is observed but it is not so this should be modified.

L198: Is there a reason why a CL only, and PE only experiment was not carried out to conclusively show CL is the preferred choice for lipid binding?

L257: LeoA was not shown yet to bind lipid so best not include the L1 loop for LeoA in your comparison.

L280: Agreed that IniA almost certainly undergoes stalk-mediated self-assembly, but as stated before please clarify that this may not be the HB2 dimer interface observed in the crystal. The manuscript currently only provides weak data for any self-assembly interface (the in vivo D333K/D337K/E402K mutant).

L.346. A line describing the refinement strategy would be welcome.

Fig 1: Should be made bigger. Everything is very small and compact.

Fig3A: additional labelling please. This goes for quite a few of the other structural panels in other figures.

Fig 4D: What lipid was used for these experiments?

Fig. 4E: Panels are very small and should be enlarged.

Fig S1: The colouring in some cases such as the red and blue is very dark so very hard to see the underlying sequence. Transparency should be increased for all colours or lighter shading chosen. I would have liked to have seen an indication of fully conserved, and similarly conserved residues to have some understanding of how related these dynamins are. At the moment conservation is seemingly indicated for some chosen residues only.

Reviewer #2:

Remarks to the Author:

The authors describe the novel structure of the mycobacterial IniA GTPase, which as such provides unique insights into the diversity of the dynamin family of proteins. In combination with biochemical analysis and membrane fusion and fission assays, this work does indeed address an important question on the fundamental mechanisms by which proteins of this family manage membrane remodeling. However, baring analysis of the structure of this protein, results reported here are at best preliminary and while representing an important first step, raise more questions than answers. Perhaps, the author could consider the following to refine this work and better address functions of this protein.

The anti-parallel HB2s (seen not in solution but only in the crystal) appear to be stabilized by symmetric electrostatic interactions. However, without further experiments, I believe this model is likely to misguide future work. The authors state that "To further test the homotypic interactions of IniA..." and are presumably referring to the HB2s interaction. If so, what is the rationale to expect that membrane binding could stabilize this interaction? Testing mutants of this interface in liposome floatation experiments could better address and validate their model. This would also help explain results from the liposome tethering assays (see below).

In describing liposome-tethering assays, the authors state that tethering is diminished with GTP,

which is not the case - it seems to be abolished. This is indeed surprisingly. However, given the small changes in light scattering and the intrinsic unreliability of these read-outs, these interpretations are not entirely convincing. The authors interpret these changes to reflect disruption of homotypic interactions on the membrane caused by GTP-binding. Does the high scattering seen in the apo state decline if GTP were to be spiked into the reaction mix? Are these representative traces? If yes, then how reproducible was this read-out and how many times were these experiments performed? Perhaps, plotting normalized means with SD could give more confidence to this read-out.

Analysis of cellular distribution of IniA-GFP could be performed better, perhaps showing a collection of cells to account for intrinsic cellular variability. This seems quite significant since the FM4-64 fluorescence itself appears quite different across the cells.

With the reported low GTPase activity, it's not surprising that IniA does not display nucleotide-dependent dimerization, since these dimers are stabilized by the transition state. ITC experiments reveal a K_d of 9 μM for GMP-PNP, which is quite high. Could this be because GMP-PNP does not adequately mimic GTP? Have the authors estimated the K_d for GTP. This should be possible considering the low rate of GTP hydrolysis.

In addition to the structural insights, that IniA causes fission is an important take-home of this manuscript. However, with data shown here, this interpretation is not so convincing. Images showing SMrTs is of low resolution making it difficult to assess protein function. Do the figures show the same field before and after protein addition? Can fission events be captured in real-time? The extent of fission is quite low (few tubes cut among many others that remain intact). There seems to be quite a bit of background sticking of the protein. Could this have contributed to such low fission, or conversely cause rupture (and not fission) of the tethers. Also, with the reported low GTPase activity, it is rather surprising that IniA is shown to causes fission. The important question then it how does it do so? Were intermediates in the fission reaction analyzed? Does membrane binding itself cause constriction or does it require GTP binding? Such analyses would be very insightful and shed light on the mechanism of membrane remodeling.

Minor comments:

The authors state that the Hinge 1a of IniA likely relies on flexible residue P300, which is perhaps incorrect since a proline generally contributes to inflexibility. Presumably, the proline is necessary to maintain the hinge in this highly bent conformation.

Small correction - K486 cited in the text should be L486.

Reviewer #3:

Remarks to the Author:

Wang et al. determined the structure of the mycobacterial dynamin-like IniA in the GTP-bound and nucleotide-free forms, which superimpose almost perfectly. IniA has a related fold to the cyanobacterial bacterial-dynamin like protein (BDLP). The authors characterize nucleotide-binding and a crystallographic dimerization interface in the helical bundle, which, however, does not appear to mediate assembly in solution. They identify a membrane-binding loop at the tip of HB2. Finally, using a supported membrane-tube assay with fluorescent lipids, they found that IniA mediates GTP-dependent membrane scission.

The structural and biochemical data appear sound, but I am concerned that the novelty of these data does not justify publication in Nature Communications. After all, the structure is very similar to that of cyanobacterial BDLP, and the membrane binding site at the tip of the HB2 is also at the expected location. The most exciting findings are the membrane fission data in Figure 5, but they

would require lots of additional analysis to yield a convincing story. Altogether, in the current form, the manuscript appears to too preliminary for Nature Communication and more suited for a more specialized structural biology journal.

Fig. 1C-D, Fig. 2C, Fig. 4E are too small. Label hinge 1a and hinge 1b.

Fig. 2a: Why is GTP stable in the crystals? It should be slowly hydrolyzed. Is GTP really bound and not GDP and some ion?

Fig. 2: K46 does not appear to be a 'gamma-phosphate' attacking residue (line 150). It rather seems to bind to the beta-phosphate, so the conclusion drawn on line 150 is curious. Additionally, it is astonishing that mutation of K49 residue has such little impact on the GTPase activity. This may indicate that the physiological GTPase activity is not achieved by wt IniA.

Fig. 3A,B: All data here show that the observed HB interface is crystallographic without functional relevance. It has been shown for several bacterial dynamin-like proteins that they act in concert with a second member, which would be IniC in mycobacteria. This could be interesting to explore (and it would be novel). For example, is there an assembly-stimulated increase in GTPase activity in the presence of both proteins, and does this involve the HB interface? Does oligomerization of IniA (possibly with IniC) follow a similar mechanisms as that of BDLP, as shown, for example, by mutagenesis? Is the dimerization interface in the G domain important for hetero-dimerization, etc.?

Fig. 3C: I do not understand what we see here. Is IniA precipitating in the absence of GTP and presence of membranes? Are the liposomes aggregating? Why does it happen only in the absence of nucleotide? What do we learn from this for the function of IniA?

Fig. 5: These are the most novel findings of the manuscript, but they need a lot more analysis to be conclusive. Are liposomes fragmented in the presence of IniA and GTP? Do other nucleotides (non-hydrolysable GTP analogues, GDP, etc) promote membrane fission? Are the mutants of Fig. 2 active in this assay? Which interfaces are involved in the fission reaction? What is the effect of adding IniC?

Typos:

P3, line 62 – The trace of helices in the stalks exits ...

Reformulate.

Minor point, The order of mention of the figures in the text does not always follow the ordering inside the figures.

Table 1: I / sigma (I), not delta.

Reviewer #4:

Remarks to the Author:

In this manuscript, Wang and colleagues have presented crystal structures of a bacterial dynamin-like protein from a Mycobacterium species in apo and GTP-bound forms. They then comprehensively characterized this protein by various biochemical assays and structural comparison with other BDLPs and human mitofusins, and revealed some special features of the Mycobacterium BDLP (mBDLP) in self-assembly, GTP hydrolysis and lipid binding. They found that Mycobacterium BDLP is able to catalyze membrane fission.

The novelty of this study includes the GTP-bound structure, the GTP-independent self-association, and membrane-fission activity of mBDLP. These data provide some advance in the dynamin field. Overall, I am enthusiastic about this study. Below are some concerns and suggestions:

1. Line 94 to 96: This sentence has problems in grammar, or a period is missing.

2. Line 118: I suggest to use "GTPase domain" rather than "GTPase". Same for the rest of the paragraph.

3. Line 164: For the AUC experiment, the authors should also test mBDLP at the transition state of GTP hydrolysis.

4. Line 165: The author mentioned that mBDLP is an "HB-stacking dimer", and then stated that the same protein is a monomer in solution. Why this is "consistently"?

5. Line 170 to 178: The authors did not prove that mBDLP can bind liposomes before starting to introduce the tethering assay. It might be more logic to prove that mBDLP can bind liposomes first.

6. Line 173 to 174: The authors found that the addition of GTP and Mg²⁺ ion caused diminishing of liposome tethering and attributed it to the GTP-loading of mBDLP. However, the effect of GTP and Mg²⁺ ion themselves was not excluded. It would be better to use ATP and Mg²⁺ ion as a negative control.

7. Still about the tethering assay. The authors stated that nucleotide-free mBDLP is a monomer in solution. They also concluded that mBDLP is not involved in membrane fusion. Therefore, it is hard to image how they mediate liposome tethering. On the other hand, the authors stated that mBDLP can tubulate liposomes and catalyze membrane fission in the presence of GTP. Would these features affect or disturb the liposome tethering results?

8. For lipid association assays, the authors are suggested to test lipid-binding of mBDLP in the presence of GMPPNP or GTPγS. The result will tell people whether and how GTP-loading affects the association between mBDLP and lipids, and may provide more information in addressing the liposome tethering assay.

9. Line 288: I suppose "they" refers to the GTPase domains of fission dynamins here. It is unclear to say they "come close" in the "tethered state". If "tethered state" means the association of the GTPase domains, then why they need to "come close" anymore? Moreover, the GTPase domains of fission dynamins do have "substantial interactions", which is essential for their function.

10. Line 289: Fission dynamin is featured by stalk-mediated polymerization, which admittedly is nucleotide-independent. However, stalk-mediated polymerization does not prevent the association between the GTPase domains (or the nucleotide-dependent association), and the latter is equally crucial for both fission and fusion dynamins. Thus, the comparison between the fission dynamins and fusion dynamins in this argument does not make much sense.

11. Following this issue, the authors concluded that mBDLP is a fission protein. Known eukaryotic fission dynamins as well as the cyanobacteria BDLP form helical polymers. Since mBDLP can tubulate liposomes as these proteins, can mBDLP also form helical polymers? If yes, according to the crystal structures, how is the polymer formed? They authors may elaborate this issue in the discussion.

12. If it is feasible, it would be intriguing to test some of the mutations in a bacterial model, which can expand the general interest of this study.

13. Figure 2C: The values in the ITC results are too small to see.

Reviewers' comments:

Reviewer #1 (Remarks to the Author):

Mycobacterial dynamin-like protein IniA mediates membrane fission: Wang et al. 2018

Wang et al. show the crystal structure of *Mycobacterium smegmatis* IniA in both the apo and GTP-bound state. IniA is a dynamin homologue and has been shown to be upregulated in the presence of key tuberculosis drug treatments and to contribute to antibiotic resistance. It is therefore an important system to work on and it is really exciting to see the crystal structure. The structure has implication both for tuberculosis drug treatment and for the dynamin field in general, which still is yet to fully clarify mechanisms of membrane fission and fusion. In principle, the structure with its coupling to membrane fission is of sufficient conceptual advance for publication in Nature Comms.

However, whilst Wang et al. are to be applauded for the many techniques they have used to dissect the *in vitro* mechanism of IniA, the manuscript is weakened by a lack of support for some of the key conjectures. Many of the obvious questions that their experiments uncover are not followed up sufficiently, or at least are not shown here. The manuscript therefore is quite limited in places for this level. Also, the discussion is weak and there are some apparent misconceptions of some dynamin biology although this may be convoluted with a language issue.

We have conducted new experiments to strengthen our understanding of the structure and function of IniA. In view of this new data, the discussion has been rewritten accordingly. Details that describe the improvements to the manuscript are presented and discussed below.

Major:

L32: The authors state that IniA associates homotypically via HB2. Whilst I don't doubt that IniA will self-assemble via HB2 in some way, there is little to support this statement in the manuscript. The only evidence is the *in vivo* D333K/D337K/E402K mutant which shows a partial effect. Does this mutant inhibit tubulation or tethering *in vitro* for example as well? This experiment would greatly support the authors hypothesis.

We have performed the *in vitro* tethering assay using the triple mutant and found little change when compared to wild type (Supplementary Fig. 4e, or see figure below). In light of these data and the fact that the *in-cell* localization defects are minor, we now believe that this interface only plays a minor role, if any in self-assembly. Therefore, we have removed the localization analysis of the triple mutant, moved Fig. 3a to supplementary Fig. 3d, and rewritten the abstract. "The IniA GTPase domain does not form detectable nucleotide-dependent dimers, but its HB2 associates homotypically" has been replaced by "IniA does not form a detectable nucleotide-dependent dimer in solution".

Also, it should be clarified that if the D333K/D337K/E402K interface is involved in self-assembly it may well be in a configuration totally different to that observed in the crystal structure with the anti-parallel HB2 association. Without some sort of supporting data the crystal contact in itself is not sufficient evidence for an assembly interface.

We now think that this interface may not be involved in self-assembly and have deleted this idea from the manuscript.

L34-36: IniA resembles mitofusin in that IniA closely resembles BDLP whose overall fold resembles that of Mitofusin 1. The actual conformation of IniA observed (hinge 1 closed) (is that what is meant by ‘configuration’?) does not closely resemble Mitofusin 1 which has not yet been shown to hinge 1 close.

We meant topology and domain structure rather than configuration. We have rephrased the sentence. It now reads “The topology and domain structure of IniA closely resemble mitofusin, another member of the dynamin superfamily that mediates mitochondrial fusion”. Additionally, the sentence in the introduction “The overall configuration of BDLP is predicted to be similar to another class of dynamin-like proteins, the mitofusins (MFNs), which mediate outer mitochondrial membrane fusion” is replaced by “The overall fold of BDLP is predicted to be similar to another class of dynamin-like proteins, the mitofusins (MFNs), which mediate outer mitochondrial membrane fusion”.

L36-38: This sentence is quite unclear and appears to be a general statement on dynamin biology. It doesn’t have much to do with the data in the manuscript for example, IniA does not show G-dimerisation. This very much weakens the abstract and the impact of the paper.

Yes, the sentence is a general statement on dynamin biology. The long-term mystery as to whether BDLP’s activity is due to fusion vs. fission, makes it worth mentioning. The proposed rule deduced from this work and other dynamin-related studies, should be informative for other scientists in the field. The sentence has been revised: “Comparison of these two dynamins suggests that their remodeling activities, fission or fusion, are related to the tendency of nucleotide-dependent or -independent association.”.

L97: Crystal structure of IniA have been obtained in both a GTP-bound and apo state. The RMSD between them is just 0.5 Å. The lack of any conformational change outside of the nucleotide binding pocket is surprising although not necessarily a concern. In some dynamin systems, the

crystallization process has ultimately determined conformation irrespective of nucleotide state (Byrnes & Sondermann PNAS 2011 and Wenger et al. Plos One 2013). The authors should comment on the total lack of conformational change observed between crystal states outside of the nucleotide binding pocket, and whether they believe the GTP state observed is physiologically relevant or whether crystallization may dominate the conformation here.

The crystal structures of IniA GTP-bound and apo states may represent the physiological soluble and stable form before decorating membrane. This is a similar conclusion to that observed for the GDP-bound and apo forms of cyanobacteria BDLP. A different conformation for the GTP-bound state in the membrane is, however, likely. Unfortunately, we have not been able to crystallize this complex in conditions that replicate a membrane bound state.

We have added some new points to the discussion: “It is surprising that IniA maintains its conformation after GTP binding in our structures. The stalk-bended conformation we observed with GTP-bound state may be physiologically relevant before membrane binding. A different conformation of the GTP-bound state in the membrane is expected. However, to date, no suitable crystallization conditions could be found that replicate the membrane environment. The actual conformation and organization of IniA during membrane remodeling requires further analysis. In any case, the GTP-hydrolysis cycle dependent conformational changes are most likely responsible for rearrangement of the potential helical filament of IniA.”

Also, does IniA crystallise in the presence of GDP? Presumably if there is no conformation change in solution between apo and GTP states then the GDP state is equivalent and crystallises? Similarly, does IniA crystallise in the presence of GTP non-hydrolysable analogues such as GMPPNP or GTP- γ -S? Presumably this gives the same crystal form as the GTP bound? It is not being asked that the authors check all these nucleotide states if they have not already done so. It is just trying to understand whether the GTP state observed with absolutely no conformational change is valid here.

We have tried to crystallize IniA in the presence of the nucleotides mentioned above but only obtained the apo form with no ligand in the nucleotide binding site. We consider that the GTP state structure exists in solution for stability before membrane binding. The conformational change may occur when IniA binds to the membrane in the presence of GTP.

L. 100 and Fig. 1. It would be helpful if new terminology for the neck and trunk domains could be avoided i.e. HB1 and HB2. These are clearly very similar structural motifs as has previously been observed for BDLP and it would be helpful for the field if new nomenclature wasn't used for each new structure.

We have revised the nomenclature in the text and figures accordingly. We labeled “HB1/Neck” and “HB2/Trunk” in figures to clearly present the name of domains and show the domain similarity with BDLP. We have kept the HBs terms in the text for the purpose of comparison with MFN which uses HBs for domain names.

L170-178: Interesting IniA tethering activity is shown in the absence of nucleotide. The addition of GTP abolishes this tethering. This assay has the potential to give some exciting insight into IniA activity and it is frustrating that the study is so limited and lacking in many obvious variations that could easily have been done at the time. For example, 1) what happens if GTP is added after a time lag once tethering has already observed. Does turbidity then decline? 2) What happens with GDP? 3) What happens with GMPPNP? The study would be greatly powered if all these experiments were cross-checked with negative stain EM to visually validate what was going on and to help build up a compelling picture of IniA mediated membrane fission. For example, where tethering is observed in the absence of nucleotide as in Fig. 3C are protein-lipid tubes observed by EM? Are these then abolished when nucleotide is added or do they never form when GTP is included from the beginning?

We have performed the tethering assay accordingly. The new results show that 1) when GTP is added after a time lag once tethering has already observed, the turbidity does decline (Fig. 4d);

2) tethering does not occur when either GDP or GMPPNP is added (Fig. 4b). These results confirm that IniA forms a nucleotide-independent association in the context of membranes, and GTP binding may cause conformational changes that interfere with a nucleotide-independent homotypic interaction.

In addition, the negative stain EM was performed to cross-check with the tethering assay. We used liposomes containing EPL (POPE 80%, *E. coli* PG 12%, *E. coli* CA 16%, POPA 4%) to achieve readily visible tubulation by EM analysis. In the apo and GDP states, long tubules are observed (Fig. 5a). In the GMPPNP state, buds and short tubules are seen (Fig. 5a). When GTP was added to preformed membrane tubules, tubules are mostly severed (Fig. 5b), indicative of GTP-dependent membrane fission.

Notably, we did not see vesicle tethering for the apo state by EM analysis which seems inconsistent with the A_{405} measurements (Fig. 4b). It is most likely because proteins/lipids used in EM analysis are of much lower concentrations, and some large aggregates may be washed off during negative staining.

L212-213 and Fig. 4E: Great to see apparent membrane localization of IniA in the cell and convincing that the GS-linker abolishes membrane localization. However, for the top row does WT mean +IniA-GFP? It would be good to see wild type cells with no IniA-GFP stained with FM4-64. This should look like GS-linker panels FM4-64 with localization in a nice red ring right? Therefore looking at the FM4-64 panel on the top row it is unclear to me why the membrane appears perturbed and punctate, and does not look like the other FM4-64 panels. Do the authors think that IniA-GFP is perturbing the membrane? Might this be contributing to the punctate localization pattern? Although not essential, does purified IniA-GFP tubulate liposomes, so is it functional? This should be discussed.

WT does mean IniA-GFP. Wild type cell expressing cytosolic GFP exhibited smooth membrane staining (Fig. 3h). It thus appears true that ectopically expressed IniA-GFP caused membrane deformation in these cells. We have added this point to the manuscript. “Notably, plasma membrane of cells transformed with wild-type IniA-GFP became rough, as indicated by membrane dye FM4-64 (Fig. 3e). In contrast, when cells expressed cytosolic GFP alone or membrane-binding defective IniA mutants, plasma membrane was rather smooth (Fig. 3f-h). These results suggest that IniA is targeted to the cell membrane by the LI loop in bacteria and can potentially remodel membranes.”.

We have also purified IniA-GFP and found that it is capable of tubulating membranes. It attaches to the membrane and possesses wild type GTPase activity.

L227: This is really exciting to see the EM IniA protein-lipid tubes in the presence of GMPPNP. Again perhaps these experiments were done but considered not important to mention, but what happens when GTP is added, or GDP? Critically, does IniA tubulate in the absence of GTP? This is important as earlier the authors state that tethering occurs in the absence of GTP so we should see some effect right? Also what happens to IniA with GTP or GDP in the absence of lipid? Presumably just particles consistent with monomer is observed (ie no self assembly) given the AUC is monomer for apo or GMPPNP state at high concentration. It would be reassuring to know these conditions had all been checked though by EM. If the authors have access to cryo-EM showing the effect of GTP on liposomes, this would really help prove this idea that IniA induces membrane fission if vesiculation is observed etc..

As mentioned above, we have performed EM analysis with IniA, liposomes, and various nucleotides. Interestingly, IniA tubulates membranes with or without nucleotides (Fig. 5a and 5b). These results suggest that IniA likely assembles on the surface of the membrane, which correlates with *in cis* oligomerization, i.e. assembly on the same membrane. Such interactions are different from what is observed in tethering assay, where membrane attached IniA likely interacts homotypically *in trans*, i.e. assembly between apposing membranes. When GTP was added, we did see signs of membrane fission (Fig. 5b), which is consistent with the results of SMrTs. We also checked protein only by EM but failed to see particles, presumably because the size of the IniA monomer is not large enough for negative staining EM. In summary, these results nicely confirm the membrane tubulation and fission activity of IniA. The results also suggest that IniA undergoes conformational changes during its cycle of GTP binding and hydrolysis.

L241-248: It is to the authors credit that they are using so many different techniques to probe the in vitro function of IniA and the use of SMrTs is compelling. However, similar to the turbidity assay and EM studies, again a little frustrating that obvious experiments such as doing a full nucleotide screen wasn't undertaken at the time and presented. For example, with the SMrT experiment, when GMPPNP is added presumably no cleavage is detected? What happens with GDP or the S50A hydrolysis deficient mutant? These are useful controls that would help power the final conclusion to the reader that IniA GTP turnover really induces membrane fission.

We have now performed the SMrTs assay with the addition of GMPPNP or GDP, and see no membrane fission (Supplementary Fig. 5c). In addition, the hydrolysis mutant (S50A/R63A) also shows no obvious fission (Supplementary Fig. 5c). These controls strengthen our conclusion that IniA mediates GTP-dependent membrane fission.

L260: This is rather weak as no dimer is described in solution. The crystal dimer may just be a crystal interface. Also, in BDLP it is both lipid and nucleotide binding required for stalk straightening (hinge 1 opening). This needs to be clarified by the authors how IniA polymerization is triggered. Does polymerisation occur just by lipid binding or is the addition of nucleotide required too?

Based on our new data, IniA tubulates membranes even in the *apo* state (Fig. 5a). We thus think that IniA oligomerization is triggered simply by lipid binding, assuming that membrane tubulation requires IniA assembly.

L281-282: 'Usually tweaks the oligomerization pattern'. This is vague and should be clarified as GTP hydrolysis is thought to induce large scale conformational changes that will significantly change the helical lattice and ultimately breaks up the filament during membrane fission.

We have rewritten the text accordingly: "GTP binding causes conformational changes, as indicated by our tethering assay; GTP hydrolysis is needed for fission, but likely induces conformational changes that would change the helical lattice, as hinted by our electron microscopy (EM) analysis of IniA-induced membrane tubules."

L282-291: 'Touching between GTPase domains.. if it exists at all'. There is compelling data in the literature for the formation of the G-dimer in many dynamin systems (Dynamin1, DNMI-L, BDLP, atlastins, GBP1 etc).

We have revised the sentence to avoid misconceptions: “In IniA, dimerization of GTPase domains may occur when assembling *in cis* on membranes.”

L289-291: ‘Fission proteins generally emphasize... nucleotide-dependent association’. This is not very clearly written so is hard to understand but as I read it the dynamin literature just does not support this statement at all. Overall, the final paragraph is not supported by the dynamin field literature and needs reconsidering.

We have rewritten this part to avoid the claim that fission protein is weak in nucleotide-dependent dimerization. We do think that the difference in nucleotide-independent association between fusion and fission proteins holds true. Now it reads like this: “Thus, fission proteins are generally capable of nucleotide-independent homotypic association, whereas fusion proteins rely mostly on nucleotide-dependent association.”

Overall, the discussion is quite limited, should be expanded and more detailed. For example, the possible nature of the protein-lipid tubes should be discussed- do the authors speculate formation of the G-dimer and BDLP-like trunk self-association or Dynamin 1-like interface 2 stalk self-association (this is closer to what is observed in the crystal packing). Or how does the low resolution IniA EM hexamer observed in Colangell et al. Mol Micro 2005 fit with their crystal structure? Also, many bacterial dynamins function as dimers (DynA in *Bacillus* and DynA/B in *Streptomyces*). It would be good if IniC in the IniA operon could be acknowledged and its potential role in activating and co-functioning with IniC discussed briefly.

We have rewritten the discussion extensively with more details, including the following points the reviewer mentioned: 1) we speculate the nature of the protein-lipid tubular structures. We think that IniA likely assembles in a dynamin 1-like interface 2 stalk self-association. We also think that the GTP cycle triggers conformational changes that rearrange helical pattern; 2) we now mention the IniA hexamer but find it difficult to assess its relevance; 3) we add speculation on the role of IniC.

Minor:

L56. Efflux pump role highly speculative

We have deleted this part to avoid controversy. The sentence now reads “IniA mediates drug tolerance⁷, tandem BDLPs DynA and DynB in the filamentous bacteria *Streptomyces* play a role in cytokinesis during sporulation^{8,10}, and LeoA in enterotoxigenic *E. coli* strain H10407 is proposed to be linked to toxin secretion^{11,12}”.

L62-68. As stated earlier, better to use BDLP terminology for neck and trunk rather than introducing new HB terminology. Writing is unclear also. It is the neck domain that connects the GTPase domain to the trunk. Dimerization is significantly mediated by the G-dimer interface as well as the paddle in BDLP. Please can this be clarified.

We have revised the nomenclature as mentioned above. We also have clarified the writing here: “The helices in the stalk firstly exit the GTPase domain and then loop back, positioning the N- and C-termini in close proximity. The cyanobacteria BDLP forms a dimer in both apo and GDP-bound states¹³. The proximal HB relative to the GTPase domain (HB1 or the “Neck”) and the distal HB (HB2 or the “Trunk”) exhibit a sharp bend. A helix-hairpin termed the “Paddle”, which is likely a transmembrane domain, is inserted in between the third and fourth helices of the trunk. Dimerization is mainly mediated by the paddle and the GTPase domain.”

L71. Spirals is technically incorrect terminology. These are helices as the radius does not decrease with each turn.

We have replaced “spirals” with “helical filament”.

L74-75. The LeoA neck does not readily superimpose on the BDLP neck. It has a slightly different conformation so the neck comparison should be avoided.

We have rewritten the sentence. It now reads “The GTPase domains can be superimposed with that of the cyanobacteria BDLP, and there is no bending in the stalk region”.

L84. Can be applied is a little strong here as this is still speculative.

We have rewritten the sentence. It now reads “The conformational changes observed in BDLP are referred by predicting models of the fusogenic action of MFN^{17,18}”.

L85-90. No role in cytokinesis has been suggested, only a speculative role in determining thylakoid morphology and cell shape. Also, it is unclear whether BDLP mediates fission or fusion reactions. So far a role in membrane fusion has speculatively been assigned due to the low nucleotide turnover rate in comparison to fission dynamins and absence of assembly stimulated turnover. At this point, suggesting a fission phenotype should be avoided. Opa1 for example generates well-ordered protein-lipid tubes and has a fusion phenotype so there is no rule of thumb that protein-lipid tube formation = fission dynamin.

We have rewritten the sentence to avoid misconceptions: “However, many other dynamin superfamily members, including dynamins themselves, also mediate membrane fission.”.

L116-117. The sentence is confusing- IniA does not form an HB open conformation. Are the authors describing BDLP here?

The HB-open state means if there was an IniA dimer, then HB1 would point in the opposite direction. We have rewritten the sentence to read “The HB1 of IniA has little contact with the GTPase domain and adopts a conformation similar to the “HB-open” state of MFN1-MGD¹⁶”.

L141. BDLP-GMPPNP has only been determined at low resolution by cryo-EM so the position of K46 should not be commented upon here.

We have rewritten this part by deleting “GMPPNP”. It now reads “This lysine is conserved in cyanobacteria BDLP, but it is not close enough to the phosphates in the GDP state.”.

L143. Please label the G2 loop on Fig. 2A. Does the U shaped flap on top of the GTP mean that the G-dimer would be inhibited from forming? The authors should clarify this as it may be important for IniA function if this flap precludes G-dimerisation.

New labels have been added to Fig. 2a. The flap points away from the potential dimer interface and thus would not interfere with dimerization. We have clarified this in the text by adding “By comparison, the equivalent G2 loop in GDP-bound cyanobacteria BDLP is largely disordered and further away from the nucleotide (Fig. 2b). Notably, the G2 of cyanobacteria BDLP has a similar shape to that of IniA, suggesting such G2s would not interfere with dimerization of the GTPase domain.”.

L168-169: The anti-parallel HB2 association observed in the crystal is not supported by the AUC. ‘Rather weak’ in solution suggests that some dimerization is observed but it is not so this should be modified.

We have modified the sentence: “These results indicate that IniA does not undergo dimerization in solution.”.

L198: Is there a reason why a CL only, and PE only experiment was not carried out to conclusively show CL is the preferred choice for lipid binding?

We had hoped to carry out these experiments, but unfortunately CL or PE alone are unable to make stable liposomes due to their own lipid-properties.

L257: LeoA was not shown yet to bind lipid so best not include the L1 loop for LeoA in your comparison.

We have changed the sentence to “In addition, the potential LI loop in LeoA and B-inserts in other DLPs point directly towards the membrane at the tip of the stalk¹¹, whereas that of IniA is parallel to HB2”.

L280: Agreed that IniA almost certainly undergoes stalk-mediated self-assembly, but as stated before please clarify that this may not be the HB2 dimer interface observed in the crystal. The manuscript currently only provides weak data for any self-assembly interface (the in vivo D333K/D337K/E402K mutant).

We have changed the text to “IniA undergoes nucleotide-independent self-assembly, which would be important for wrapping around a constricted membrane area.”.

L346: A line describing the refinement strategy would be welcome.

The sentence has been revised to read “Refinement was carried out using PHENIX allowing the XYZ coordinates to vary and in real-space. Each atom was refined to have an individual isotropic B-factor. The rigid body motion of the protein was fitted according to TLS parameters”.

Fig 1: Should be made bigger. Everything is very small and compact.

The revised figure is shown below.

Fig3A: additional labelling please. This goes for quite a few of the other structural panels in other figures.

We have added labels accordingly.

Fig 4D: What lipid was used for these experiments?

E. coli polar lipids (EPL). This has now been indicated in the figure legend: “**b** Liposome (*E. coli* polar lipids containing 78 mol% PE, 12 mol% PG, 6 mol% CL, 4 mol% PA) flotation assay showing the effect of NaCl concentration (150 mM and 600 mM) on IniA membrane

association.”, “**d** As shown in (b), but with mutants at the LI loop. $\Delta 480-492$, L480-K492 was replaced by “GGGGSGGGGS” linker.”

Fig. 4E: Panels are very small and should be enlarged.

The panels have now been enlarged.

Fig S1: The colouring in some cases such as the red and blue is very dark so very hard to see the underlying sequence. Transparency should be increased for all colours or lighter shading chosen. I would have liked to have seen an indication of fully conserved, and similarly conserved residues to have some understanding of how related these dynamins are. At the moment conservation is seemingly indicated for some chosen residues only.

The coloring has been modified to make it easier to see the sequences. Due to the inclusion of MFN, which is important for remodeling comparison, conserved residues are largely those from the G-motifs.

INIA_MYCSM	-----MIVELIDHTSAIAAAKDR-----ADL	21
INIA_MYCTU	MVPAGLCAYRDLRRKRARKWGDVTQPDDPRRVGVIVELIDHTIAIAKLNER-----GDL	55
BDLP_NOSP7	MVNQVATDRFIQDLERVAQVRSEMSVCLNKL---AETINKAELAGDSSSGKLSLER--D	54
MFNI_HUMAN	MAEFPVSLKHFVLAKKAITAIFDQLLEFVTEGSHFVEATYKNPELDRIAIEDDLVEMQGY	60
INIA_MYCSM	VERLRAAKARISDPQIRVVVIAAGQLKQGKSQLLNLSLNIIPVARVGDDESTVLAIVVSYGEQ	81
INIA_MYCTU	VQRLTRARQRITDPQVRVVIAGLQKQKSQLLNLSLNLPAARVGDDEATVVIIVVSYSAQ	115
BDLP_NOSP7	IEDITIASKNLQGVFRLLVLGDMKRKSTFLNALIENLLPSDVNPCTAVLITVLRYPGE	114
MFNI_HUMAN	KDKLSIIGEVLSRRHMKVAFFGRTSSGKSSVINAMLWDKVLPSGIGHITNCFLSVEGTDG	120
INIA_MYCSM	ASARLVVVARPDGAEPETLIPSEVTTDLRRAPQASGRQV-----LRVEVTAAPSPL	132
INIA_MYCTU	PSARLVLAAGPDGTTAAVDIPVDDISTDVRRAPHAGGREGV-----LRVEVGAAPSPL	166
BDLP_NOSP7	KKVTIHFNDG---KSPQQLDFQNFYKYTIDPAEAKKLEQEKKQAFPDVDYAVVEYPLTL	171
MFNI_HUMAN	DKAYLMTESGD--EKKSVKTVNQLAHALHMDKDLKAGC-----LVRVFWKAKC	167
INIA_MYCSM	LK--GGLAFVITPGVGGHGQPHLSATLGLLPDADAMLMSDTSQEFTEPEMKFIR-QALE	189
INIA_MYCTU	LR--GGLAFITPGVGGGLGQPHLSATLGLLPEADAVLVVSDTSQEFTEPEMWFVR-QAHQ	223
BDLP_NOSP7	LQ--KGTIEIVISPLGN-DTEARNELSLGYVNNCHAILFVMRASQPCITLGERRYLENYIKG	228
MFNI_HUMAN	ALLRDDLVLVISPGTDVITELDSWIDKFCL-DADVFLVANSESTLMNTEKHFPHKVNER	226
INIA_MYCSM	IC-PVAAIVATKTDLYPHWR-QIVDANIAHLQRA-----LNVP	226
INIA_MYCTU	IC-PVGAVVATKTDLYPHWR-EIVNANAHLQAR-----VMP	260
BDLP_NOSP7	RG-LTVFVLNVAWQVRESLIDPDVVEELQASENRLR-QVFANLAIEYCTVEGQNIYDER	286
MFNI_HUMAN	LSKPNIFILNRRWASASEPEYMEDVRRQHMERCLHFLV-----EELKVVNALEAQR	279
INIA_MYCSM	VIPASSVLSRSHASLNDKELNEESN----FP---AIVKFLSEHVLSRQ-----T	268
INIA_MYCTU	IIAVSSLRSHAVTLNDKELNEESN----FP---AIVKFLSEQVLSRA-----T	302
BDLP_NOSP7	VFELSSIQALRRRLKNPQADLDGTG----FPKFMDSLNTFLTRERIAE-----L	332
MFNI_HUMAN	IFFVSAKEVLSARKQKAQGMPEGSVALAEGFHARLQEFQNFQIFEECISQSAVKTKEFQ	339
INIA_MYCSM	DRIRDQIVDEIRSAAEHLLAVESELSSFNDPGERER-----LTAELERRKQEAQDALQQ	323
INIA_MYCTU	ERVVAGVLGEIRSATEQLAVSLGSELVVNDPNLRDR-----LADLERRKRKREAAQAVQQ	357
BDLP_NOSP7	RQVRTLARLACNHTREAVARRIPLL-----EQDVNELKKRIDSVEPEFNKLTGIRDEFQK	387
MFNI_HUMAN	HTIRAKQILATVKNIMDSVNLAA-----EDKRHSVEEREDQIDRLDFIRNQMLL	390
INIA_MYCSM	TALWQQVLSDGIADLTADVHDLRHFRFRIAAHTEKVIDGCDPTLH-----WAEIGAE	376
INIA_MYCTU	TALWQQVLDGFDLTADVHDLRTRFRFTVTEAERQIDSCDPTAH-----WAEIGND	410
BDLP_NOSP7	EIINTRDQARTISESFRSYVLNLGNTFENDFLRYQPELNLDFLSSGKREAFNAALQKA	447
MFNI_HUMAN	TLDVKKIKIKEVTEEVANKVSCAMTDEICRLSVLVD EFCSEFHPNPDLVKIYKSELNKHIE	450
INIA_MYCSM	LEDAVATAVGDNFVWAYQRAEALAAEVARTEAAGLDAVQMPQIDARMDGAGFGELNSLA	436
INIA_MYCTU	VENAIAATAVGDNFVWAYQRSEALADDVARSFADAGLDSVLSAELSPHVMGTDFGRKALG	470
BDLP_NOSP7	FEQYITDKSAAWTLTAEKDINAAPKELSRASQYGASYNQITDQITEKLTGKDVKVHVT	507
MFNI_HUMAN	DGMGRNLADRCTDEVNALVLTQQEIENLKPILPAGIQDKLHTLIPCCKFDLSYNLNYH	510
INIA_MYCSM	RLEAKPIKIGHKVVTGMRGSYGGVLMFGMLTSFAGL-----	473
INIA_MYCTU	RMESKPLRRGHKMIIGRGSYGGVVMIGLSSVVGLG-----	507
BDLP_NOSP7	TAEEDNSPGWAKWAMGLLSLSKGNLAGFALAGAGFD-----	543
MFNI_HUMAN	KLCSDFQEDIVFPFSLGWSSLVHRELGPRNAQRVLLGLSEPIQLPRSLASTPTAPTTPA	570
INIA_MYCSM	-----MFNPLSLGAGFVLGRKAYKEDMENRMLRVRNEAKAN	509
INIA_MYCTU	-----LNFPLSVGAGLILGRMAYKEDKNRLLRVRSEAKAN	543
BDLP_NOSP7	---WKNILLNYFTVIGIGGITAVTGILLGPIGFALLGLGVGLQADQARRELVKTKAKE	600
MFNI_HUMAN	TPDNASQEELMITLVGLASVTSRTSMGIIIVGGVIWKTIGWKLSSVSLTMYGALYLYER	630
INIA_MYCSM	VRFVDDVAFVVGKESRDRLKGIQRQLRDHYREIANQTTLSLNSLQAAIAAAKVEEAE	569
INIA_MYCTU	VRRFVDDISFVVSKQSRDLKMIQRLLRDHYREIAEETRSLTESLQATIAAAQVAETER	602
BDLP_NOSP7	LVKHLPQVAHEQSQVYVNAVKECFDSYEREVSKRINDDIVSRKSELNDLNLVKQKQREINR	660
MFNI_HUMAN	LSWTTTHAKERAFKQQFVNYATEKLRMIVSSTSANCSHQVKQIATTFARLCQQVDITQKQ	690
INIA_MYCSM	NTRVRELERQONILKQVVDHAAKLAQDPAPSPA-----	602
INIA_MYCTU	DNRIRELQRQLGILSQVNDNLAGEPTLTPRASLGRA-----	640
BDLP_NOSP7	ESEFNRLKNLQEDVIAQLQKIEAAYSNLLAYSS-----	693
MFNI_HUMAN	LEEEIARLPKEIDQLEKIQNNSKLLRNKAVQLENELNFTKQFLPSSNEES	741

Reviewer #2 (Remarks to the Author):

The authors describe the novel structure of the mycobacterial IniA GTPase, which as such provides unique insights into the diversity of the dynamin family of proteins. In combination with biochemical analysis and membrane fusion and fission assays, this work does indeed address an important question on the fundamental mechanisms by which proteins of this family manage membrane remodeling.

However, barring analysis of the structure of this protein, results reported here are at best preliminary and while representing an important first step, raise more questions than answers. Perhaps, the author could consider the following to refine this work and better address functions of this protein.

We have now improved this study by conducting more experiments, and undertaking further analysis to provide a better understanding of the functions of IniA. Details that describe the improvements to the manuscript are presented and discussed below.

The anti-parallel HB2s (seen not in solution but only in the crystal) appear to be stabilized by symmetric electrostatic interactions. However, without further experiments, I believe this model is likely to misguide future work. The authors state that "To further test the homotypic interactions of IniA..." and are presumably referring to the HB2s interaction. If so, what is the rationale to expect that membrane binding could stabilize this interaction? Testing mutants of this interface in liposome floatation experiments could better address and validate their model. This would also help explain results from the liposome tethering assays (see below).

We have tried the floatation assay and tethering assay with the D333K/D337K/E402K triple mutant, and failed to observe differences when compared to the wild type (Supplementary Fig. 4e). In light of these results, we have removed the localization results for the triple mutant and moved the illustration of the HB2 interface to supplementary Fig. 3d. We have also modified the abstract and discussion accordingly. The tethering assay and the tubulation assay indicate that homotypic association of IniA does exist, but the interface remains to be identified.

In describing liposome-tethering assays, the authors state that tethering is diminished with GTP, which is not the case - it seems to be abolished. This is indeed surprisingly. However, given the small changes in light scattering and the intrinsic unreliability of these read-outs, these interpretations are not entirely convincing. The authors interpret these changes to reflect disruption of homotypic interactions on the membrane caused by GTP-binding. Does the high scattering seen in the apo state decline if GTP were to be spiked into the reaction mix? Are these representative traces? If yes, then how reproducible was this read-out and how many times were

these experiments performed? Perhaps, plotting normalized means with SD could give more confidence to this read-out.

We now included a visual assay to support the conclusion of the absorbance-based tethering assay (Fig. 4c). IniA-mediated vesicle tethering is reversible, as addition of GTP causes a decrease in previously established A_{405} readings (Fig. 4d). To answer the last part of the question, we have plotted mean values with SD error bars to confirm that the absorbance readings are statistically valid (Supplementary Fig. 4d).

Analysis of cellular distribution of IniA-GFP could be performed better, perhaps showing a collection of cells to account for intrinsic cellular variability. This seems quite significant since the FM4-64 fluorescence itself appears quite different across the cells.

We have improved the quality of these images (Fig. 3e-h). These are shown below.

With the reported low GTPase activity, it's not surprising that IniA does not display nucleotide-dependent dimerization, since these dimers are stabilized by the transition state. ITC experiments reveal a K_d of $9 \mu\text{M}$ for GMP-PNP, which is quite high. Could this be because GMP-PNP does not adequately mimic GTP? Have the authors estimated the K_d for GTP. This should be possible considering the low rate of GTP hydrolysis.

As suggested, we have now performed ITC analysis with IniA and GTP. The K_d value is $0.74 \mu\text{M}$ for GTP (Fig. 2c) whilst for GMPNP the K_d value is $9 \mu\text{M}$. These results confirm that GMPNP does not adequately mimic GTP. Similar results have been reported for ATL, Sey1p and MFN1, where subtle atomic changes can be sensed by this class of GTPases. The lack of nucleotide-

dependent dimerization of IniA is less likely due to varied nucleotide affinity, because even the addition of GTP does not induce dimerization (Fig. 4a).

In addition to the structural insights, that IniA causes fission is an important take-home of this manuscript. However, with data shown here, this interpretation is not so convincing. Images showing SMrTs is of low resolution making it difficult to assess protein function. Do the figures show the same field before and after protein addition? Can fission events be captured in real-time?

We now include a series of time-lapse images (Fig. 5d) and a movie (Supplementary Movie 1) to show fission events.

The extent of fission is quite low (few tubes cut among many others that remain intact). There seems to be quite a bit of background sticking of the protein. Could this have contributed to such low fission, or conversely cause rupture (and not fission) of the tethers.

As mentioned above, we now provide fission images with improved quality (Fig. 5c and Supplementary Fig. 5c). We think that the low efficiency of fission is attributed to low GTPase activity of IniA.

Also, with the reported low GTPase activity, it is rather surprising that IniA is shown to cause fission. The important question then is how does it do so? Were intermediates in the fission reaction analyzed? Does membrane binding itself cause constriction or does it require GTP binding? Such analyses would be very insightful and shed light on the mechanism of membrane remodeling.

We now provide additional controls to show that fission is GTP hydrolysis dependent (Fig. 5c and Supplementary Fig. 5c). A GTPase-defective mutant S50A/R63A attached efficiently to membrane tubes, but failed to mediate fission in the presence of GTP (Supplementary Fig. 5c).

Minor comments:

The authors state that the Hinge 1a of IniA likely relies on flexible residue P300, which is perhaps incorrect since a proline generally contributes to inflexibility. Presumably, the proline is necessary to maintain the hinge in this highly bent conformation.

We have modified the text accordingly: “Both hinge 1a and hinge 1b are shaped by a cluster of charged residues (Supplementary Fig. 2a)”.

Small correction - K486 cited in the text should be L486.

We have corrected this typo.

Reviewer #3 (Remarks to the Author):

Wang et al. determined the structure of the mycobacterial dynamin-like IniA in the GTP-bound and nucleotide-free forms, which superimpose almost perfectly. IniA has a related fold to the cyanobacterial bacterial-dynamin like protein (BDLP). The authors characterize nucleotide-binding and a crystallographic dimerization interface in the helical bundle, which, however, does not appear to mediate assembly in solution. They identify a membrane-binding loop at the tip of HB2. Finally, using a supported membrane-tube assay with fluorescent lipids, they found that IniA mediates GTP-dependent membrane scission.

The structural and biochemical data appear sound, but I am concerned that the novelty of these data does not justify publication in Nature Communications. After all, the structure is very similar to that of cyanobacterial BDLP, and the membrane binding site at the tip of the HB2 is also at the expected location. The most exciting findings are the membrane fission data in Figure 5, but they would require lots of additional analysis to yield a convincing story. Altogether, in the current form, the manuscript appears to too preliminary for Nature Communication and more suited for a more specialized structural biology journal.

Base on the feedback from all four reviewers, we have significantly improved the paper by conducting more experiments, thereby allowing a greater depth of analysis and discussion to explain and describe the function of IniA. The implications of this study have a broader impact as they expand the overall knowledge of the functionality of the dynamin superfamily.

Fig. 1C-D, Fig. 2C, Fig. 4E are too small. Label hinge 1a and hinge 1b.

We have made these panels larger and labelled the hinges.

Fig. 2a: Why is GTP stable in the crystals? It should be slowly hydrolyzed. Is GTP really bound and not GDP and some ion?

GTP hydrolysis by IniA virtually stops at 16 °C, which is the crystallization temperature. This makes it possible to obtain the structure with GTP bound. The density between β - and γ -phosphate groups is continuous, which suggests the gamma phosphate is bonded covalently to complete the structure of GTP (Fig. 2a).

Fig. 2: K46 does not appear to be a ‘gamma-phosphate’ attacking residue (line 150). It rather seems to bind to the beta-phosphate, so the conclusion drawn on line 150 is curious.

The ϵ -amino group of K46 points towards the linkage between the β - and γ -phosphate with its terminal nitrogen closer to γ -phosphate (2.9 Å) than to β -phosphate (3.6 Å) (Fig. 2a). In addition, ITC analysis reveals that the K46A mutant has a similar affinity for GDP as the wild-type enzyme and there is only a two-fold reduction in affinity for GMPPNP or GTP (Fig. 2c). Thus, the role of K46 appears to be β - γ bond attacking instead of β -phosphate binding.

Additionally, it is astonishing that mutation of K49 residue has such little impact on the GTPase activity. This may indicate that the physiological GTPase activity is not achieved by wt InIA.

We were also surprised by this result. The lysine at this position usually contributes to nucleotide binding. Wild type purified InIA exhibits equivalent affinity for GTP and GDP similar to other DLPs. Other residues in nucleotide-binding site exhibit reasonable effects in GTPase activity assay. Besides, we also observed GTP-hydrolysis dependent membrane fission. We thus believe that the GTPase activity we measured is of physiological relevance.

Fig. 3A,B: All data here show that the observed HB interface is crystallographic without functional relevance. It has been shown for several bacterial dynamin-like proteins that they act in concert with a second member, which would be InIC in mycobacteria. This could be interesting to explore (and it would be novel). For example, is there an assembly-stimulated increase in GTPase activity in the presence of both proteins, and does this involve the HB interface? Does oligomerization of InIA (possibly with InIC) follow a similar mechanisms as that of BDLP, as shown, for example, by mutagenesis? Is the dimerization interface in the G domain important for hetero-dimerization, etc.?

We agree and now confirm that the HB2 interface only plays a minor role. We have removed the localization results for the D333K/D337K/E402K triple mutant and moved the illustration of the HB2 interface to supplementary Fig. 3d. We have also modified the abstract and discussion accordingly.

Unfortunately, purified IniC is very unstable, making the proposed analysis technically impossible. Nevertheless, we speculate the involvement of IniC in the discussion as suggested.

Fig. 3C: I do not understand what we see here. Is IniA precipitating in the absence of GTP and presence of membranes? Are the liposomes aggregating? Why does it happen only in the absence of nucleotide? What do we learn from this for the function of IniA?

We show that IniA binds to membranes with the same affinity in the presence or absence of nucleotides. The tethering assay presented here indicates that IniA forms homotypic interactions *in trans*, i.e. bridging membranes. Such tethering is confirmed by the new additional visual assay (Fig. 4c). These results show that IniA can self-associate, at least in a nucleotide-independent manner.

Fig. 5: These are the most novel findings of the manuscript, but they need a lot more analysis to be conclusive. Are liposomes fragmented in the presence of IniA and GTP? Do other nucleotides (non-hydrolysable GTP analogues, GDP, etc) promote membrane fission? Are the mutants of Fig. 2 active in this assay? Which interfaces are involved in the fission reaction? What is the effect of adding IniC?

We have now performed assays with additional controls. Membrane can be tubulated by IniA, with or without nucleotides (Fig. 5a and 5b). Only when GTP is added are liposomes fragmented (Fig. 5b) and membrane fission occurs (Supplementary Fig. 5c) Additionally, S50A/R63A binds to membranes but fails to cut them, even when GTP is added (Supplementary Fig. 5c). The interface involved in fission reaction is yet to be identified, and we are unable to obtain stable IniC.

Typos:

P3, line 62 – The trace of helices in the stalks exits ...

Reformulate.

We have rewritten the sentence: “The helices in the stalk firstly exit the GTPase domain and then loop back, positioning the N- and C-termini in close proximity.”.

Minor point, The order of mention of the figures in the text does not always follow the ordering inside the figures.

We have now placed the links to the figures in consecutive order.

Table 1: I / sigma (I), not delta.

We have corrected this.

Reviewer #4 (Remarks to the Author):

In this manuscript, Wang and colleagues have presented crystal structures of a bacterial dynamin-like protein from a *Mycobacterium* species in apo and GTP-bound forms. They then comprehensively characterized this protein by various biochemical assays and structural comparison with other BDLPs and human mitofusins, and revealed some special features of the *Mycobacterium* BDLP (mBDLP) in self-assembly, GTP hydrolysis and lipid binding. They found that *Mycobacterium* BDLP is able to catalyze membrane fission.

The novelty of this study includes the GTP-bound structure, the GTP-independent self-association, and membrane-fission activity of mBDLP. These data provide some advance in the dynamin field. Overall, I am enthusiastic about this study.

Below are some concerns and suggestions:

1. Line 94 to 96: This sentence has problems in grammar, or a period is missing.

We have modified the text: “To gain insight into IniA-mediated drug resistance, we determined the structures of full-length IniA from *Mycobacterium smegmatis* (*M. smegmatis*), which shares 68% sequence identity with IniA from *Mycobacterium tuberculosis* (*M. tuberculosis*) (Supplementary Fig. 1).”.

2. Line 118: I suggest to use “GTPase domain” rather than “GTPase”. Same for the rest of the paragraph.

We have modified the text accordingly.

3. Line 164: For the AUC experiment, the authors should also test mBDLP at the transition state of GTP hydrolysis.

We have performed AUC analysis as suggested. The addition of GDP-AIF₄⁻ did not induce dimerization of IniA (Fig. 4a).

4. Line 165: The author mentioned that mBDLP is an “HB-stacking dimer”, and then stated that the same protein is a monomer in solution. Why this is “consistently”?

We have modified the text by deleting “consistently”. It now reads “To assess dimer formation in solution, we performed analytical ultra-centrifugation (AUC). Purified IniA behaved as a monomer and remained so in the presence of GTP, GDP, GMPPNP, or GDP-AIF₄⁻ (Fig. 4a).”.

5. Line 170 to 178: The authors did not prove that mBDLP can bind liposomes before starting to introduce the tethering assay. It might be more logic to prove that mBDLP can bind liposomes first.

We have switched the order of Fig. 3 and 4 and adjusted the text as suggested.

6. Line 173 to 174: The authors found that the addition of GTP and Mg²⁺ ion caused diminishing of liposome tethering and attributed it to the GTP-loading of mBDLP. However, the effect of GTP and Mg²⁺ ion themselves was not excluded. It would be better to use ATP and Mg²⁺ ion as a negative control.

We have added this negative control as suggested. The disruption of IniA-mediated tethering is less likely due to charge alteration by Mg²⁺ and the nucleotides, because the addition of Mg²⁺ and ATP did not affected tethering (Fig. 4b and 4c).

7. Still about the tethering assay. The authors stated that nucleotide-free mBDLP is a monomer in solution. They also concluded that mBDLP is not involved in membrane fusion. Therefore, it is hard to image how they mediate liposome tethering. On the other hand, the authors stated that mBDLP can tubulate liposomes and catalyze membrane fission in the presence of GTP. Would these features affect or disturb the liposome tethering results?

The tethering results are clear and reproducible. Given that nucleotide binding does not affect membrane association of IniA, it is highly probable that when bound to membranes, IniA forms homotypic interactions *in trans*, i.e. bridging liposomes. Such interactions can only occur in the absence of nucleotides. In the membrane tubulation assay, we believe that we are looking at potential IniA self-assembly *in cis*, i.e. on the same membrane.

8. For lipid association assays, the authors are suggested to test lipid-binding of mBDLP in the presence of GMPPNP or GTPgammaS. The result will tell people whether and how GTP-loading affects the association between mBDLP and lipids, and may provide more information in addressing the liposome tethering assay.

Additional controls (GTP, GDP, GMPPNP, ATP) have now been included (Supplementary Fig. 4c). The results suggest that IniA lipid binding is not affected by nucleotide states.

9. Line 288: I suppose “they” refers to the GTPase domains of fission dynamins here. It is unclear to say they “come close” in the “tethered state”. If “tethered state” means the association of the GTPase domains, then why they need to “come close” anymore? Moreover, the GTPase domains of fission dynamins do have “substantial interactions”, which is essential for their function.

The reviewer may have misinterpreted our meaning of “they”. It actually refers to the stalk regions of fusion proteins. The text has been modified to make clearer expression: “During this process, their stalk regions usually rely on the motion triggered by the GTPase domain. The stalk regions may then come close to each other in the tethered state. However, they often lack specific substantial interactions in between.”.

10. Line 289: Fission dynamin is featured by stalk-mediated polymerization, which admittedly is nucleotide-independent. However, stalk-mediated polymerization does not prevent the association between the GTPase domains (or the nucleotide-dependent association), and the latter is equally crucial for both fission and fusion dynamins. Thus, the comparison between the fission dynamins and fusion dynamins in this argument does not make much sense.

We have modified the text to avoid this misconception: “Thus, fission proteins are generally capable of nucleotide-independent homotypic association, whereas fusion proteins rely mostly on nucleotide-dependent association.”.

11. Following this issue, the authors concluded that mBLDP is a fission protein. Known eukaryotic fission dynamins as well as the cyanobacteria BDLP form helical polymers. Since mBDLP can tubulate liposomes as these proteins, can mBLDP also form helical polymers? If yes, according to the crystal structures, how is the polymer formed? They authors may elaborate this issue in the discussion.

We expected that IniA would form helical polymers on membranes, but did not see any interpretable pattern by EM. The HB2 interface is proven less relevant, which has not been supported by our latest investigation. It is thus difficult to speculate how IniA assembly occurs at this point. We have now commented on the issue of the IniA assembly in the discussion.

12. If it is feasible, it would be intriguing to test some of the mutations in a bacterial model, which can expand the general interest of this study.

We would like to test IniA function by setting up an effective bacterial model, however, this is a major undertaking, which we would like to do in a follow up publication.

13. Figure 2C: The values in the ITC results are too small to see.

We have modified the figure.

Reviewers' Comments:

Reviewer #1:

Remarks to the Author:

Mycobacterial dynamin-like protein IniA mediates membrane fission

Wang et al. have made extensive and overall positive improvements to the manuscript, which is now much more robust. Here they show the novel crystal structure of *Mycobacterium smegmatis* IniA in both apo and GTP conformation. They show that it both binds and turns over nucleotide. They nicely identify the lipid binding domain of IniA at the trunk tip and show that it binds lipid in vitro. They show that IniA is located on the membrane in vivo. IniA tubulates membrane in the presence and absence of nucleotides and this effect is somehow correlated with liposome aggregation through an unclear mechanism – possibly by tethering or a consequence of tubulation. The addition of GTP diminishes liposome aggregation and in the case of the SMrT assay compellingly induces membrane fission. Despite there being a lot of data, the actual interesting and really novel part of this manuscript is the SMrT assay and the first apparently clear sign of membrane fission in a bacterial dynamin. The crystal structure is unfortunately very similar to BDLP so does not add much novelty. The manuscript for me is quite unclear in how it is written, and weak in places in how the data is interpreted. For example, I still don't understand this key idea in the abstract L39-41 and later on in L349-357 as it appears not readily supported by current literature (please see below*). As this is presented in the abstract as the conclusion of the paper I find this weakens the manuscript. Overall, the IniA structure is an important addition to the dynamin field, and it is exciting to see membrane fission. However, the broad conclusions and the lack of clarity in the writing leaves me feeling borderline as to whether the manuscript, as it is now, quite has the quality and impact for publication in Nature Comms.

*My understanding of what is being suggested is as follows: that IniA fission dynamin forms a nucleotide independent filament/polymer on the membrane, and that Mitofusin fusion dynamin only self-associates in the presence of nucleotide (G-dimerisation is suggested) and not by their cytosolic domains (L350-351). But this ignores the fact that the mitofusins are unlikely to exist as monomers on the membrane with current data suggesting they are tetramers (Ishihara et al., J. Cell Science 2004). Any dimer or tetramer would be expected to include cytosolic domain interfaces for oligomerization. So if the authors mean mitofusins don't readily form filaments or polymers then that should be clarified (however, they may ultimately form filaments or short polymeric assemblies and this just hasn't yet been captured due to the difficulty of purification and handling). Additionally, the mechanism of membrane fusion suggested for Mitofusin is currently just one of two. The other relies on mitofusin polymerization and disassembly (Daumke and Roux, Current Biology, 2017). All these papers seem to suggest the hypothesis presented here at this time (or at least how it is written) is unsafe/premature. Finally, the architecture of any IniA filament is also completely unknown. Lipid tethering and tubulation may just be a consequence of protein crowding (unlikely but possible at this point- the authors show no high mag images of the tubes with filamentous repeats shown/layer lines in the fourier transform). For me, there is not sufficient data presented to support this rule of thumb IniA fission vs Mitofusin fusion membrane remodeling theory at this time.

Reviewer #2:

Remarks to the Author:

The revised manuscript now satisfactorily address all the points raised in the previous round of review. I recommend publication of this manuscript.

Reviewer #3:

Remarks to the Author:

The manuscript and its figures have been significantly improved by the revision, but the story is still not conclusive, e.g. I am not convinced that IniA is a membrane scission molecule akin to dynamin. Considering this, also the general statements on membrane fission and fusion proteins are way too speculative and may be entirely misleading. I believe this work still requires more functional experiments to make a conclusive story and to be a candidate for Nature Communications.

Major:

1.) Function of IniA in membrane remodeling.

The authors show in Figure 4 that IniA tethers vesicles in the nucleotide-free state suggesting that it is a membrane tethering molecule. In Figure 5, the authors show that IniA cleaves membranes in a GTP-dependent fashion. I do not know how to reconcile these apparently opposing observations, suggesting that IniA is a tethering and fission molecule at the same time? Very likely, one (or both) of the in vitro assays results in an artificial outcome, and it is unclear which one it is. In the absence of at least some functional cell biological data, I am generally skeptical to transfer observations from in vitro membrane remodeling assays to physiological functions, since the in vitro assays are prone to artefacts.

What is IniA doing in these spot-like plasma membrane assemblies seen in Figure 3e that look very similar to those formed by BDLP in *N. punctiforme*? Producing intracellular vesicles by membrane fission? I am not aware that such vesicles exist in Mycobacteria and that a GTPase-dependent membrane scission reaction as seen for dynamin makes any sense in this context.

2.) From a structural point of view, the authors convincingly show that IniA is a close relative of mitofusin (and BDLP), a well characterized membrane fusion protein. With this in mind and having removed now any additional mutagenesis-based information on relevant assembly interfaces, I find it difficult to believe that IniA could have a similar helical assembly as dynamin featuring a criss-cross arrangement of the HBs. I find it also hard to believe that it has a similar mechanism of action, e.g. a nucleotide-hydrolysis driven membrane constriction mechanism, since this crucially involves the formation of helical filaments and interfilament contacts via the GTPase domain. Thus, in the absence of a stimulated GTPase activity and likely the absence of a G interface, a nucleotide-hydrolysis dependent constriction mechanism as in dynamin is very unlikely. With the current structural data, it seems more probable that IniA assembles on membranes in a similar fashion as BDLP and mitofusin and has a similar mechanism of action as these two proteins – and IniC may be involved in such mechanism. It may create membrane curvature as a prerequisite for some membrane remodeling activity (which can be fusion or scission or stabilization of membrane curvature, see for example, Fig. 7 in Low et al, Cell 2009). At least some of the candidate assembly interfaces in the HB based on the oligomerized BDLP structure should be biochemically tested to see whether mutations in them affect assembly and membrane tubulation to make a convincing claim.

Criticism to the existing experiments:

Fig. 5a: The tubule diameters should be properly quantified for all conditions from a range of tubules.

Fig. 5c/d: The fragmentation data should be quantified, showing the number of examined tubules, the number of breaking points per tubule, etc.

Fig. 5d: When looking at the slow time course of these experiments, IniA appears to be a very inefficient membrane scission protein, at least compared to dynamin. Again, this sheds doubts on the relevance of the observed membrane cleavage. Maybe what we observe in this assay is just a GTP-dependent tight assembly of IniA on membranes, leading at some point to breakage of the membrane? As the author show, GMPPNP does not properly mimic GTP binding in IniA, and the examined GTP hydrolysis mutants may also not be able to assemble in the same fashion as wt IniA

with GTP. Does GTPgammaS bind with the same affinity as GTP to IniA and induce the same membrane scission phenotype?

Minor:

K46: Is this residue at the same position in the P-loop as the catalytic arginine in GBP1/atlastin? If yes, this may be worth mentioning.

Fig. 1d: Better show rates here instead of reaction velocities.

Fig. 5d: Should these tubules not retract when cleaved? Why do they stay straight?

Reference list: Ref 16 and 17 are duplicated – probably Cao, Meng, Chen et al. was meant instead?

Typos:

line 199 flotation or floatation?

line 732 wildtype

Reviewer #4:

Remarks to the Author:

The authors have fully addressed my concerns. I therefore recommend acceptance of this manuscript.

Minor:

Line 350: "but does not show nucleotide-independent association of the cytosolic domain"---better specify that this is "trans" association.

Reviewers' comments:

Reviewer #1 (Remarks to the Author):

Mycobacterial dynamin-like protein IniA mediates membrane fission

Wang et al. have made extensive and overall positive improvements to the manuscript, which is now much more robust. Here they show the novel crystal structure of *Mycobacterium smegmatis* IniA in both apo and GTP conformation. They show that it both binds and turns over nucleotide. They nicely identify the lipid binding domain of IniA at the trunk tip and show that it binds lipid in vitro. They show that IniA is located on the membrane in vivo. IniA tubulates membrane in the presence and absence of nucleotides and this effect is somehow correlated with liposome aggregation through an unclear mechanism – possibly by tethering or a consequence of tubulation. The addition of GTP diminishes liposome aggregation and in the case of the SMrT assay compellingly induces membrane fission. Despite there being a lot of data, the actual interesting and really novel part of this manuscript is the SMrT assay and the first apparently clear sign of membrane fission in a bacterial dynamin. The crystal structure is unfortunately very similar to BDLP so does not add much novelty. The manuscript for me is quite unclear in how it is written, and weak in places in how the data is interpreted. For example, I still don't understand this key idea in the abstract L39-41 and later on in L349-357 as it appears not readily supported by current literature (please see below*). As this is presented in the abstract as the conclusion of the paper I find this weakens the manuscript. Overall, the IniA structure is an important addition to the dynamin field, and it is exciting to see membrane fission. However, the broad conclusions and the lack of clarity in the writing leaves me feeling borderline as to whether the manuscript, as it is now, quite has the quality and impact for publication in Nature Comms.

*My understanding of what is being suggested is as follows: that IniA fission dynamin forms a nucleotide independent filament/polymer on the membrane, and that Mitofusin fusion dynamin only self-associates in the presence of nucleotide (G-dimerisation is suggested) and not by their cytosolic domains (L350-351). But this ignores the fact that the mitofusins are unlikely to exist as monomers on the membrane with current data suggesting they are tetramers (Ishihara et al., J. Cell Science 2004). Any dimer or tetramer would be expected to include cytosolic domain interfaces for oligomerization. So if the authors mean mitofusins don't readily form filaments or polymers then that should be clarified (however, they may ultimately form filaments or short polymeric assemblies and this just hasn't yet been captured due to the difficulty of purification and handling). Additionally, the mechanism of membrane fusion suggested for Mitofusin is currently just one of two. The other relies on mitofusin polymerization and disassembly (Daumke and Roux, Current Biology, 2017). All these papers seem to suggest the hypothesis presented here at this time (or at least how it is written) is unsafe/premature. Finally, the architecture of any IniA filament is also completely unknown. Lipid tethering and tubulation may just be a consequence of protein crowding (unlikely but possible at this point- the authors show no high mag images of the tubes with filamentous repeats shown/layer lines in the fourier transform). For me, there is not sufficient data presented

to support this rule of thumb IniA fission vs Mitofusin fusion membrane remodeling theory at this time.

We thank Rev #1 for their overall comments on our revised manuscript. In particular, the comment that the SMrT assay is novel and the data give the first apparently clear sign of membrane fission in a bacterial dynamin.

We also appreciate the comment “Overall, the IniA structure is an important addition to the dynamin field, and it is exciting to see membrane fission.” We fully agree that the structural data is novel in that GTP-bound state of BDLP family proteins has not been reported previously at high resolution, and the lipid-interacting loop in the dynamin superfamily has not been visualized until now.

In L39-41 and later on in L349-357, we stated that fusion or fission activity by dynamin superfamily can be estimated by their dimerization/oligomerization tendency. In accord with the last point, we have removed the statements in L39-41 and L349-357, and have rewritten the text to discuss fission vs. fusion rather than suggest specific rules.

One point we would like to clarify is that when we say nucleotide-independent oligomerization, we mean the cytosolic regions only (see the third paragraph of Discussion section). We agree with the reviewer that MFN, and also ATL, forms nucleotide-independent self-association using the TM segments. However, there is no evidence to show that the stalk (HB) regions of MFN form homo-oligomers. We have added this point in the revision.

Reviewer #2 (Remarks to the Author):

The revised manuscript now satisfactorily address all the points raised in the previous round of review. I recommend publication of this manuscript.

Reviewer #3 (Remarks to the Author):

The manuscript and its figures have been significantly improved by the revision, but the story is still not conclusive, e.g. I am not convinced that IniA is a membrane scission molecule akin to dynamin. Considering this, also the general statements on membrane fission and fusion proteins are way too speculative and may be entirely misleading. I believe this work still requires more functional experiments to make a conclusive story and to be a candidate for Nature Communications.

Major:

1.) Function of IniA in membrane remodeling.

The authors show in Figure 4 that IniA tethers vesicles in the nucleotide-free state suggesting that it is a membrane tethering molecule. In Figure 5, the authors show that IniA cleaves membranes in a GTP-dependent fashion. I do not know how to reconcile these apparently opposing observations, suggesting that IniA is a tethering and fission

molecule at the same time? Very likely, one (or both) of the *in vitro* assays results in an artificial outcome, and it is unclear which one it is. In the absence of at least some functional cell biological data, I am generally skeptical to transfer observations from *in vitro* membrane remodeling assays to physiological functions, since the *in vitro* assays are prone to artefacts.

What is IniA doing in these spot-like plasma membrane assemblies seen in Figure 3e that look very similar to those formed by BDLP in *N. punctiforme*? Producing intracellular vesicles by membrane fission? I am not aware that such vesicles exist in Mycobacteria and that a GTPase-dependent membrane scission reaction as seen for dynamin makes any sense in this context.

The tethering assay shows that IniA can associate *in trans*, confirming that it is capable of forming homotypic interactions. The membrane tubulation assay and fission assay suggest that IniA probably forms homotypic interactions *in cis*. Thus, these two observations could happen at the same time, both of which are reconcilable and meaningful. Plasma membrane deformation seen in Fig. 3E, likely been tubulated/evaginated membranes, resembles membrane remodeling mediated by dynamin. The enrichment of IniA at these sites is consistent with its homotypic assembly on membranes. These data also suggest that IniA is an intrinsically weak fission dynamin. At this stage, it is impossible to pin point the exact role of IniA. We speculate in our discussion that such fission activity can be used in membrane repair, as IniA inducing drugs are known to damage the plasma membrane. Nevertheless, it is still important to clarify the molecular activity of BDLP.

Since the reviewer asked “if IniA produces intracellular vesicles by membrane fission”, we performed a 3D SIM reconstruction using PM dye in IniA-overexpressed cells. The experiments were carried out under the same conditions as in Fig. 3E. The internalization of the FM4-64 signal was captured, which strongly supports the concept of membrane fission by IniA (Supplementary Fig. 6).

Since the reviewer also requested functional cell biological data, we tested IniA mutants in a drug resistance assay using *M. smegmatis* and validated the effects of key residues on protein function *in vivo* (the new Fig. 6). These data have been added to the text.

2.) From a structural point of view, the authors convincingly show that IniA is a close relative of mitofusin (and BDLP), a well characterized membrane fusion protein. With this in mind and having removed now any additional mutagenesis-based information on relevant assembly interfaces, I find it difficult to believe that IniA could have a similar helical assembly as dynamin featuring a criss-cross arrangement of the HBs. I find it also hard to believe that it has a similar mechanism of action, e.g. a nucleotide-hydrolysis driven membrane constriction mechanism, since this crucially involves the formation of helical filaments and interfilament contacts via the GTPase domain. Thus, in the absence of a stimulated GTPase activity and likely the absence of a G interface, a nucleotide-hydrolysis dependent constriction mechanism as in dynamin is very unlikely. With the current structural data, it seems more probable that IniA assembles on membranes in a similar fashion as BDLP and mitofusin and has a similar mechanism of action as these two proteins – and IniC may be involved in such mechanism. It may create membrane curvature as a prerequisite for some membrane remodeling activity (which can be fusion or scission or stabilization of membrane curvature, see for example, Fig. 7 in Low et al, Cell 2009). At least some of the candidate assembly interfaces in the HB based on the oligomerized BDLP structure should be biochemically tested to see whether mutations in them affect assembly and membrane tubulation to make a convincing claim.

As mentioned in our discussion, we agree with the reviewer that IniA is less likely to adopt a mechanism similar to dynamin, and more likely to assemble as occurs for cyanobacteria BDLP. It is also very likely that IniC is involved. However, the EM structure of assembled cyanobacteria BDLP is relatively low resolution. More importantly, even with very similar overall shapes, the sequences of IniA and cyanobacteria BDLP are poorly conserved. It is thus impossible to speculate on assembly interfaces without a high-resolution structure of assembled IniA. Thus, we removed the HB2/Trunk of IniA and tested the mutant in membrane remodeling assays. The results showed that the truncated IniA binds to membranes and possesses wild type GTPase activity, but fails to tether liposomes *in trans* and fails to tubulate and cut membranes

(Fig.2d, Supplementary Fig.4f, 4g, 5e). These data suggest that nucleotide-independent association of IniA is needed and likely involves the HB2/Trunk domain.

Criticism to the existing experiments:

Fig. 5a: The tubule diameters should be properly quantified for all conditions from a range of tubules.

We have now performed these measurements (Supplementary Fig. 5c).

Fig. 5c/d: The fragmentation data should be quantified, showing the number of examined tubules, the number of breaking points per tubule, etc.

We have now performed this quantification (Supplementary Fig. 5b).

Fig. 5d: When looking at the slow time course of these experiments, IniA appears to be a very inefficient membrane scission protein, at least compared to dynamin. Again, this sheds doubts on the relevance of the observed membrane cleavage. Maybe what we observe in this assay is just a GTP-dependent tight assembly of IniA on membranes, leading at some point to breakage of the membrane? As the author show, GMPPNP does not properly mimic GTP binding in IniA, and the examined GTP hydrolysis mutants may also not be able to assemble in the same fashion as wt IniA with GTP. Does GTPgammaS bind with the same affinity as GTP to IniA and induce the same membrane scission phenotype?

We agree that the fission activity of IniA is not comparable to dynamin. Nevertheless, it clearly exists and is GTP-dependent. Even if it is due to GTP-dependent tight assembly, it is still considered as a fission event. The K_d of GMPPNP for IniA is $\sim 9 \mu\text{M}$. When performing SMrT experiments, the nucleotide concentration is 2 mM, which would be sufficient to allow full occupancy of IniA. Different from GTP, GTPgammaS was shown to bind very weakly to IniA or the K46A mutant (see figures below), thus we were unable to do further analysis using GTPgammaS.

Minor:

K46: Is this residue at the same position in the P-loop as the catalytic arginine in GBP1/atlastin? If yes, this may be worth mentioning.

Yes, it is equivalent to R77 of human atlastin 1 (ALT1, another dynamin-like GTPase that mediates fusion of the endoplasmic reticulum). We have added this point in the text.

Fig. 1d: Better show rates here instead of reaction velocities.

The Fig. 2d has been revised.

Fig. 5d: Should these tubules not retract when cleaved? Why do they stay straight?

The tubes are attached to PEG-treated glass. The same fission pattern was seen when dynamin-1 was used to demonstrate the methods.

Reference list: Ref 16 and 17 are duplicated – probably Cao, Meng, Chen et al. was meant instead?

We have corrected these references.

Typos:
line 199 flotation or floatation?

We have corrected the text.

line 732 wildtype

We have corrected the text.

Reviewer #4 (Remarks to the Author):

The authors have fully addressed my concerns. I therefore recommend acceptance of this manuscript.

Minor:

Line 350: "but does not show nucleotide-independent association of the cytosolic domain"---better specify that this is "trans" association.

We have modified the text.

Reviewers' Comments:

Reviewer #1:

Remarks to the Author:

Thank you to the authors who have invested a lot of effort to improve the manuscript from the first revision. It is now much improved and may be published once the following minor comments have been addressed:

L. 69. The BDLP paddle only mediates dimerization in the GDP state.

L.234. Please mention that the stacking dimer is observed as crystal packing for clarity.

L.236. There is no evidence that the HB2-stacking interface as observed in the crystal is involved here, only that the HB2/trunk domain is required for tethering.

L.337. Personally I don't agree with this. IniA structure is super similar to BDLP so I still think it more likely to self-assemble like BDLP rather than stalk cross-over.

L.357. Is reference 24 correct here? It doesn't appear relevant. Given the structure of IniA is compared to other BDLPs it would be relevant to mention the recently solved *Campylobacter* DLP1/DLP2 structures (Liu et al. Nature Comms 2018). These were also observed to tether membrane, so again some mention of these in the context of IniA membrane tethering may be useful in the relevant place in the manuscript, perhaps L365. The *Campylobacter* DLP1/DLP2 structures are also evidence for BDLP hetero-oligomerisation and supports the idea of potential IniA and IniC interaction L370.

L.379. change 'the helical lattice' to 'any helical lattice' as there does not appear to be any firm evidence for a helical polymer as of yet.

L382-389 should be referenced better (e.g L383) and could be better written. It is still rather unclear to read. L384-385 is not supported as mitofusins likely exist as tetramers on the membrane (Ishihara et al., 2004). L389 in particular is unclear/ambiguous to me.

Reviewer #3:

Remarks to the Author:

The manuscript has been further improved by additional experiments and changes in the text and is now suited for publication in Nature Communications. However, there is one more important point for the authors to address. In addition, some editing of the English language may still be required, especially in the discussion. A few suggestions and text edits for the consideration of the authors are included below.

Main point:

Supplementary Figure 5c and line 336: Average diameter of membrane tubules in the presence and absence of GDP is 20 and 30 nm and the comparison of the diameter to the BDLP tubules. Something appears wrong here, 20 nm diameter is just too thin for a membrane tubule covered with a regular dynamin-like protein coat. A thin membrane tubule has an outer diameter of 5 nm, then there are just 75 Angstrom left on both sides for the oligomeric IniA coat. The thinnest outer diameter of an oligomeric dynamin-like membrane assembly that I am aware of is 40 nm for the dynamin in the superconstricted state.

Are the tubules observed here at all coated with protein? Or are they protein aggregates? The authors need to carefully check this not to mislead the reader.

Line 143 – equivalent to the catalytic R77 of human atlastin

Line 164 – confirming the role of K46 in bond breakage.

The evidence for K46 as a catalytic residue is still weak since its mutation merely reduces GTP hydrolysis (not more than the affinity, for example). Better remove this statement.

L307 wrinkeless – better wrinkles

L310 Membrane invaginations and

L311 whether the membrane remodeling activity

L330 – exhibits a GTPase and a helical domain,

L349 – The bent conformation of the stalk we observed in the GTP-bound state is ...

L352 – which, however, cannot be crystallized in the absence of membranes.

L366 IniA-IniC – The recent structural paper from Harry Low (Structural basis for membrane tethering by a bacterial dynamin-like pair) in Nature Communications has to be included in this discussion. In fact, the study is very relevant for this manuscript.

Line 394 – remodels the plasma membrane

L860 – The diagram shows the statistical analysis of results presented in Fig. 5c.

Reviewers' comments:

Reviewer #1 (Remarks to the Author):

Thank you to the authors who have invested a lot of effort to improve the manuscript from the first revision. It is now much improved and may be published once the following minor comments have been addressed:

L. 69. The BDLP paddle only mediates dimerization in the GDP state.

Response: The sentence has been revised like this: “Dimerization is mainly mediated by the paddle and the GTPase domain in the GDP state”.

L.234. Please mention that the stacking dimer is observed as crystal packing for clarity.

Response: The sentence has been revised like this: “However, we discovered no such interface but a HB2-stacking dimer in the crystal packing of our structures”.

L.236. There is no evidence that the HB2-stacking interface as observed in the crystal is involved here, only that the HB2/trunk domain is required for tethering.

Response: We have changed the statement like this: “Symmetric interactions of a D333^{α7}-R351^{α7} salt bridge and D337^{α7}-H344^{α7}, E402^{α8}-H348^{α7} hydrogen bonds are observed between the anti-parallel HB2s”.

L.337. Personally I don't agree with this. IniA structure is super similar to BDLP so I still think it more likely to self-assemble like BDLP rather than stalk cross-over.

Response: It has been revised.

L.357. Is reference 24 correct here? It doesn't appear relevant. Given the structure of IniA is compared to other BDLPs it would be relevant to mention the recently solved Campylobacter DLP1/DLP2 structures (Liu et al. Nature Comms 2018). These were also observed to tether membrane, so again some mention of these in the context of IniA membrane tethering may be useful in the relevant place in the manuscript, perhaps L365. The Campylobacter DLP1/DLP2 structures are also evidence for BDLP hetero-oligomerisation and supports the idea of potential IniA and IniC

interaction L370.

Response: The reference 24 has a sequence alignment with secondary structure annotation in Fig. S1. The arrangement of secondary structure of MFN1 is similar to IniA, thus it should be referenced. As the reviewer suggested, we have also mentioned *Campylobacter* DLP1/DLP2 for tethering and hetero-oligomerization in the suggested place in the manuscript.

L.379. change 'the helical lattice' to 'any helical lattice' as there does not appear to be any firm evidence for a helical polymer as of yet.

Response: It has been revised.

L382-389 should be referenced better (e.g L383) and could be better written. It is still rather unclear to read. L384-385 is not supported as mitofusins likely exist as tetramers on the membrane (Ishihara et al., 2004). L389 in particular is unclear/ambiguous to me.

Response: It has been revised.

Reviewer #3 (Remarks to the Author):

The manuscript has been further improved by additional experiments and changes in the text and is now suited for publication in Nature Communications. However, there is one more important point for the authors to address. In addition, some editing of the English language may still be required, especially in the discussion. A few suggestions and text edits for the consideration of the authors are included below.

Main point:

Supplementary Figure 5c and line 336: Average diameter of membrane tubules in the presence and absence of GDP is 20 and 30 nm and the comparison of the diameter to the BDLP tubules.

Something appears wrong here, 20 nm diameter is just too thin for a membrane tubule covered with a regular dynamin-like protein coat. A thin membrane tubule has an outer diameter of 5 nm, then there are just 75 Angstrom left on both sides for the

oligomeric IniA coat. The thinnest outer diameter of an oligomeric dynamin-like membrane assembly that I am aware of is 40 nm for the dynamin in the superconstricted state.

Are the tubules observed here at all coated with protein? Or are they protein aggregates? The authors need to carefully check this not to mislead the reader.

Response:

We thank the reviewer for pointing this out. We speculate that IniA fails to form a regular pattern on the membrane during EM analysis. Therefore, the measured diameters do not include the layer of proteins. To confirm that membrane deformation requires the presence of protein on the membrane, we performed GUV-based experiments using fluorescence labeled lipids and proteins. When the same lipid composition (EPL: 78 mol% phosphatidylethanolamine [PE]; 12 mol% phosphatidylglycine [PG]; 6 mol% cardiolipin [CL]; and 4 mol% phosphatidic acid [PA]) as the EM analysis was used, GUV is poorly generated (extremely low yield and in very small sizes). When the PE percentage was reduced (40 mol% PG, 39 mol% POPE, 10 mol% CA, 8 mol% PI, 2 mol% DOPS, and 1 mol% rhodamine-PE), GUV was readily formed. We did not see tubule formation when IniA was incubated with GUV. However, the membrane formed buds/puncta and some clouds, similar to deformed plasma membrane seen in IniA-overexpressed cells. We reasoned that the lack of tubule formation is likely due to lower levels of PE in the membrane. In this context, we were able to show membrane deformation by IniA requires its physical presence on the membrane. More convincingly, in the GUV-based assay, we observed GTP-dependent rupture of the membrane by IniA, further strengthening the argument that it is a scission DLP. To avoid confusion, we have now removed the EM results and replaced it with the GUV-based results.

Line 143 – equivalent to the catalytic R77 of human atlastin

Response: It has been revised.

Line 164 – confirming the role of K46 in bond breakage.

The evidence for K46 as a catalytic residue is still weak since its mutation merely reduces GTP hydrolysis (not more than the affinity, for example). Better remove this statement.

Response: The statement “confirming the role of K46 in β - γ bond breakage” has been removed.

L307 wrinkleless – better wrinkles

Response: It has been revised.

L310 Membrane invaginations and

Response: It has been revised.

L311 whether the membrane remodeling activity

Response: It has been revised.

L330 – exhibits a GTPase and a helical domain,

Response: It has been revised.

L349 – The bent conformation of the stalk we observed in the GTP-bound state is ...

Response: It has been revised.

L352 – which, however, cannot be crystallized in the absence of membranes.

Response: It has been revised.

L366 IniA-IniC – The recent structural paper from Harry Low (Structural basis for membrane tethering by a bacterial dynamin-like pair) in Nature Communications has to be included in this discussion. In fact, the study is very relevant for this manuscript.

Response: We have mentioned Harry Low et al work for IniA-IniC discussion and referenced their paper.

Line 394 – remodels the plasma membrane

Response: It has been revised.

L860 – The diagram shows the statistical analysis of results presented in Fig. 5c.

Response: It has been revised.

Reviewers' Comments:

Reviewer #3:

Remarks to the Author:

The new GUV assays are nicely contributing to the functional characterization of IniA in membrane remodeling. The paper is ready for publication now. A few suggestions for text improvements that I DO NOT need to see again are still indicated below.

L 69 – remove mainly ? (or are there other contributions) ?

Line 130 adopts a very similar fold

Line 144 – points towards the bridging oxygen between the beta- and gamma-phosphates

Line 146 but it does not contact the phosphates in the GDP-bound state

Line 148 – likely stabilizing the positions of ...

Line 175 – remove 'with'

Line 180 many dynamin superfamily members interact with membranes

Line 203 – Analyzing the lipid binding mode, we ...

Line 219 which is predominantly on the plasma membrane, occasionally in a punctate pattern

Line 223 – presumably indicating that this mutant protein aggregates ...

L224 and 227 – 'the plasma membrane'

Line 232 – nucleotide-dependent dimer formation via the GTPase domain ...

Line 251 – visualized tethering by fluorescent microscopy

Line 271 – and occasionally remodeled it into buds (I am not sure what is meant with 'cloud' here), reminiscent ...

Fig. 5d – labelling of y-axis is missing

Line 280 – is lost the right word here ? Maybe better: broken / ruptured

Line 280, 281, line 282, –GUVs (not GUV)

Line 309 ... and Supplementary Fig. 5b). In this case, ... (remove 'only').

Line 338 – nucleotide-dependent dimers in solution

Line 349, 350 – it forms almost flat hexamers

Line 399 the cytosolic domains of ATL and MFN do not show nucleotide-dependent association (I guess that is what is meant? However, is the cytosolic domain of MFN not a monomer and that of ATL anyhow a GTPase-dependent dimer? Is this comparison really helpful here?).

REVIEWERS' COMMENTS:

Reviewer #3 (Remarks to the Author):

The new GUV assays are nicely contributing to the functional characterization of IniA in membrane remodeling. The paper is ready for publication now. A few suggestions for text improvements that I DO NOT need to see again are still indicated below.

L 69 – remove mainly ? (or are there other contributions) ?

Response: It has been revised.

Line 130 adopts a very similar fold

Response: It has been revised.

Line 144 – points towards the bridging oxygen between the beta- and gamma-phosphates

Response: It has been revised.

Line 146 but it does not contact the phosphates in the GDP-bound state

Response: It has been revised.

Line 148 – likely stabilizing the positions of ...

Response: It has been revised.

Line 175 – remove 'with'

Response: It has been revised.

Line 180 many dynamin superfamily members interact with membranes

Response: It has been revised.

Line 203 – Analyzing the lipid binding mode, we ...

Response: It has been revised.

Line 219 which is predominantly on the plasma membrane, occasionally in a punctate pattern

Response: It has been revised.

Line 223 – presumably indicating that this mutant protein aggregates ...

Response: It has been revised.

L224 and 227 – 'the plasma membrane'

Response: It has been revised.

Line 232 – nucleotide-dependent dimer formation via the GTPase domain ...

Response: It has been revised.

Line 251 – visualized tethering by fluorescent microscopy

Response: It has been revised.

Line 271 – and occasionally remodeled it into buds (I am not sure what is meant with 'cloud' here), reminiscent ...

Response: It has been revised.

Fig. 5d – labelling of y-axis is missing

Response: It has been revised.

Line 280 – is lost the right word here ? Maybe better: broken / ruptured

Response: It has been revised.

Line 280, 281, line 282, –GUVs (not GUV)

Response: It has been revised.

Line 309 ... and Supplementary Fig. 5b). In this case, ... (remove 'only').

Response: It has been revised.

Line 338 – nucleotide-dependent dimers in solution

Response: It has been revised.

Line 349, 350 – it forms almost flat hexamers

Response: It has been revised.

Line 399 the cytosolic domains of ATL and MFN do not show nucleotide-dependent association (I guess that is what is meant? However, is the cytosolic domain of MFN not a monomer and that of ATL anyhow a GTPase-dependent dimer? Is this comparison really helpful here?).

Response: We have deleted this sentence.